# A 1985-2023 time series dataset of absolute reservoir storage in Mainland Southeast Asia (MSEA-Res)

Shanti Shwarup Mahto[1,3*], Simone Fatichi[1], and Stefano Galelli[2]

[1]Department of Civil and Environmental Engineering, National University of Singapore, 117576, Singapore
[2]School of Civil and Environmental Engineering, Cornell University, Ithaca, NY 14853, USA
[3]Department of Geoinformatics, Central University of Jharkhand, Ranchi, 835222, India

*Correspondence to: Shanti Shwarup Mahto (ssmahto.clim@gmail.com, ss.mahto@nus.edu.sg)

## Abstract

The recent surge in reservoir construction has increased global surface water storage, with Mainland Southeast Asia (MSEA) being a significant hotspot. Such infrastructural evolution demands updates in water management strategies and hydrological models. However, information on actual reservoir storage is hard to acquire, especially for transboundary river basins. To date, no high spatio-temporal dataset on absolute storage time series is available for reservoirs in MSEA. To address this gap, we present (1) a comprehensive, open-access database of absolute storage time series (sub-monthly) for 186 reservoirs (larger than 0.1 km$^3$) in MSEA spanning the period 1985-2023, and (2) an analysis of the reservoir storage dynamics. This dataset is derived from remote sensing observations, integrating satellite-based water surface area extraction from high-resolution (30m) images and Area-Elevation-Storage (AES) relationships to estimate reservoir level and storage dynamics. The MSEA-Res database includes static (Area-Elevation-Storage curves, water frequency, reservoir extent) and dynamic (area, water level, and absolute storage time series) components for each reservoir. The 186 reservoirs collectively store around 175 km³ of water, with a minimum of 140 km³ and a maximum of 210 km³. They cover an average area of 8,700 km², ranging from a minimum of 6,500 km² to a maximum of 10,000 km². We show that the combined average reservoir storage has increased from 70 km³ to 160 km³ (+130%) from 2008 to 2017, primarily contributed by reservoirs in the Irrawaddy, Red, Upper Mekong, and Lower Mekong basins. Our in-situ validation provides a good match between estimated storage and in-situ observations, with 50% of the validation sites (10 out of 20) showing an $R^2 > 0.7$ and an average nRMSE < 14%. The indirect validation (based on altimetry-converted storage) shows even better results, with an $R^2 > 0.7$ and an average nRMSE < 12% for 70% (14 out of 20) of the reservoirs. Furthermore, the analysis of the 2019-2020 drought event in the MSEA region reveals that nearly 30-40% of the region experienced more than five months of drought, with the most significant impact on reservoirs in Cambodia and Thailand. As a result, storage departures ranged up to -40% in some reservoirs, highlighting significant impacts on water availability. Overall, this analysis demonstrates the potential of the inferred storage time series for assessing real-life water-related problems in Mainland Southeast Asia, with the possibility of applying the method to estimate reservoir storage time series in other parts of the world. The reservoir storage database in Mainland Southeast Asia (MSEA-Res database) and the associated Python code are publicly available on Zenodo at https://zenodo.org/records/14844580 (Mahto et al., 2025).

## 1 Introduction

Water reservoirs cause some of the most significant human-induced alterations of the hydrological cycle, influencing the distribution of water in space and time (Chao et al., 2008; Cooley et al., 2021; Haddeland et al., 2014; Lehner et al., 2011). The construction of reservoirs can also lead to significant environmental and socio-economic impacts, including biodiversity loss, alterations of geochemical cycles, and changes in land use patterns (Degu et al., 2011; Kirchherr et al., 2016; Maavara et al., 2020; Vörösmarty et al., 2010; Winemiller et al., 2016). Despite these impacts, reservoirs remain pivotal in generating renewable energy and supporting water management, thus driving the demand for new reservoirs (Chao et al., 2008; Wada et al., 2017), especially in the Global South. Accurate information on reservoir operations is thus crucial for practitioners, policymakers, and scientists to estimate water budgets, assess hydrological and nutrient fluxes, project water availability for hydropower generation, and mitigate flood and drought risks (Bakken et al., 2014; Chao et al., 2008). Information on the temporal evolution of *absolute* reservoir storage, or level, is particularly useful, since it provides a direct measurement of the total volume of water stored in a reservoir at any given time – this contrast against *relative* storage time series, which only track changes in storage across a given time interval.

Currently, information on long-term absolute reservoir storage is limited across most of the globe, with consolidated datasets available only for a handful of countries (Li et al., 2023; Steyaert et al., 2022; Steyaert and Condon, 2024). Such type of information is particularly needed in regions – like Mainland Southeast Asia – that are experiencing rapid hydropower development. Laos, for instance, is realising its vision of becoming the "Battery of Asia" by constructing new hydropower reservoirs and exporting electricity to neighbouring countries; it is expected, moreover, that several additional reservoirs will become operational in the years to come (Ang et al., 2024). Similarly, other Southeast Asian countries, such as Vietnam and Cambodia, have also built most of their reservoirs in the past two decades (Ang et al., 2024; Zhang and Gu, 2023), altering the flow of transboundary rivers and raising tensions between countries. With this concern in mind, we focus on the reservoirs of Mainland Southeast Asia, including Myanmar, Thailand, Laos, Vietnam, Cambodia, Malaysia, Singapore, and part of southern China—where several major rivers originate and flow through the region.

The problem of inferring reservoir storage time series can only be partially addressed with the aid of hydrological models, since some basic information on operational strategies – typically not available – is needed to setup and validate models (Dang et al., 2020, p.2; Galelli et al., 2022; Hanasaki et al., 2006; Nazemi and Wheater, 2015a, b; Vu et al., 2022; Wada et al., 2017). Fortunately, advances in remote sensing offer a viable opportunity to estimate storage by relating information on reservoir surface area and elevation (Busker et al., 2019; Gao et al., 2012; Tortini et al., 2020; Vu et al., 2022). For this task, information on reservoir bathymetry – synthesized by Area-Elevation-Storage (A-E-S) curves – becomes crucial. It is indeed common practice to derive the A-E-S curves from remotely-sensed digital elevation models (DEMs) (Zhang and Gao, 2020); their time of acquisition, however, may limit the available information. When the DEM captures the reservoir's topography before its

filling begins, absolute storage estimation is possible using these remotely sensed data (Li et al., 2023). For reservoirs constructed before the DEM was made available, the problem lies in the fact that satellite-based DEMs do not provide information below the reservoir water surface, leading to a partially unknown bathymetry. However, even in such cases, remotely sensed water surface area from sensors like Landsat, Sentinel, and MODIS, or water level data from satellite
altimeters such as JASON, Sentinel-6, and SARAL-Altika, can still be used to estimate storage changes (Das et al., 2022; Minocha et al., 2024; Zhang et al., 2014). Therefore, while recent studies have quantified long-term surface area and storage changes of reservoirs at global (Busker et al., 2019; Hou et al., 2024; Tortini et al., 2020) and regional scales (Shen et al., 2023; Song et al., 2022), absolute storage estimations – especially for those reservoirs built before 2000 (acquisition year of SRTM DEM) – are still uncertain in space and time because of the lack of detailed bathymetry information (Hao et al., 2024; Li et al.,
2023; Zhang and Gao, 2020).

One potential approach to improving such estimates is radar altimetry, which has proven useful for measuring water levels in lakes and reservoirs (Markert et al., 2019; Schwatke et al., 2015; Vu et al., 2022). Yet, limited coverage is a major limitation in popular altimetry-based datasets, such as Hydroweb (Crétaux et al., 2011), G-REALM (Birkett et al., 2011), and DAHITI
(Schwatke et al., 2015). For Mainland Southeast Asia, altimetry-based water level data are available for only a few (20-30) reservoir overpasses. On top of that, the available water level datasets are not continuous in time. Although time series datasets are available for reservoir storage anomaly (Shen et al., 2022, 2023), none of them provide long-term time series for absolute reservoir storage. Some studies modelled total storage – only for a few reservoirs –  using LiDAR data (Bacalhau et al., 2022; Chen et al., 2022; Li et al., 2020), surrounding topographical information (Fang et al., 2023; Liu et al., 2020; Liu and Song,
2022), or through simplified modelling approaches (Khazaei et al., 2022; Yigzaw et al., 2018). However, they show inaccuracies in storage estimates for reservoirs that were built before 2000, because of the (necessary) assumptions on reservoir bathymetry (Hao et al., 2024; Li et al., 2023; Zhang and Gao, 2020). Other studies relied on field surveys to create three-dimensional (3D) bathymetry maps to estimate absolute storage, but these are limited to very few reservoirs (Busker et al., 2019; Weekley and Li, 2019).

Recently, the GloLakes database was produced by Hou et al., (2024), providing absolute water storage dynamics for lakes from 1984 to present - for 27,000 global lakes and reservoirs using the geostatistical model described in Messager et al., (2016). Although Hou et al., (2024) covers the entire globe by providing a comprehensive dataset for large-scale assessments, it has a few limitations for the reservoirs located in Mainland Southeast Asia. First, the model parameters (used in the storage
estimation) strongly depend on mean depth (extrapolating the surrounding topographical slope towards the centre of the lake to estimate lake depth), the surface area of the lake (derived from Landsat satellite images), and average slope (derived from DEM). Therefore, uncertainties in the estimates of reservoir storage may be generated by the estimation of depth, slope, and through other model coefficients. Second, GloLakes does not include some of the largest reservoirs in MSEA, including

Nuozhadu (22 km$^3$), Xiaowan (15 km$^3$), Xe Kaman 1 (4 km$^3$), and Lower Seasan 2 (6 km$^3$), which play a significant role in water redistribution and hydropower generation (Ang et al., 2024; Galelli et al., 2022; Vu et al., 2022).

Here, we address these gaps by presenting a robust and comprehensive sub-monthly time series dataset of absolute reservoir storage for Mainland Southeast Asia (hereafter, "MSEA-Res database"), whose reservoir network is described in Section 2. Specifically, our open-access database includes sub-monthly time series data of absolute storage for 186 reservoirs (larger than 0.1 km$^3$) in mainland Southeast Asia (MSEA), covering the period from 1985 to 2023. The creation of this database is facilitated by two technical advances (Section 3), namely (1) the concomitant use of Landsat and Sentinel-2 images, and (2) the creation of hypsometric curves based on the new database introduced by Hao et al., (2024), which provides bathymetry information for all reservoirs in the GRanD database. The first advancement is aimed to increase the temporal resolution of our time series, while the latter allows us to address the (aforementioned) challenges concerning the estimation of hypsometric curves for reservoirs that were not recently built. To demonstrate the usefulness of MSEA-Res, we conduct a multi-basin analysis of the dynamics and trends of reservoir (absolute) storage, offering insights into how storage patterns have evolved over the years and across different basins (Section 4). Finally, we analyse the impact of the 2019-2020 drought in Mainland Southeast Asia on surface water storage, highlighting the significant effects of extreme dry weather events on water resources in Mainland Southeast Asia. Through these examples, we show that MSEA-Res can be used for a variety of applications, such as hydrological modelling, drought analyses, and regional water resources planning.

## 2 Water reservoirs in Mainland Southeast Asia

### 2.1 Dam design attributes

We first analysed global and regional reservoir databases to compile a list of reservoirs (with storage larger than 0.1 km$^3$) built in Mainland Southeast Asia until 2023. As shown in Table 1, we used two global databases [GRanD Version-1.3 (Lehner et al., 2011) and GDAT (Zhang and Gu, 2023)] and one regional database for the Mekong (Ang et al., 2024). The most popular global dam database – GRanD – was used to get the list of georeferenced reservoirs that were built until 2016. Unfortunately, the GRanD database has not been updated for post-2017 reservoirs in our study region. Therefore, we collected the list of georeferenced reservoirs built between 2017 and 2023 from more recent databases. For the Mekong basin, we used the reservoir database prepared by Ang et al., (2024), whereas the GDAT database was used for the other basins (i.e., Chao Phraya, Red, Salween, Irrawaddy, and remaining smaller river basins). Information on each reservoir in the final list of 186 elements was verified and validated against high-resolution Google Earth images. Among the reservoir attributes, we collected four main ones: name of the reservoir, spatial coordinates of the reservoir (i.e., longitude and latitude), storage capacity, and year of commission.

**Table 1. List of global and regional reservoir databases used to collect the dam design attributes.**

| Category | Database | Region | Number of reservoirs | Period | Source |
|---|---|---|---|---|---|
| Global | GRanD v1.3 (https://ln.sync.com/dl/bd47eb6b0/anhxaikr-62pmrgtq-k44xf84f-pyz4atkm/view/default/447819520013/) | Mainland Southeast Asia | 126 | Until 2016 | Lehner et al., 2011 |
| | Global Dam Tracker (GDAT) (https://www.nature.com/articles/s41597-023-02008-2/) | Mainland Southeast Asia, except the Mekong | 22 | 2017-2023 | Zhang and Gu, 2023 |
| Regional | Reservoirs in the Mekong (https://essd.copernicus.org/articles/16/1209/2024/) | Mekong basin | 38 | 2017-2023 | Ang et al., 2024 |

## 2.2 Distribution and evolution of reservoirs

Based on the acquired information, we first present the distribution and temporal evolution of reservoirs in Mainland Southeast Asia (Fig. 1). Among the 186 large reservoirs in MSEA, 125 (~68 %) were built in the twenty-first century. As a result, a dense network of newly constructed reservoirs spreads across all basins, with the exception of the Chao Phraya, western Lower Mekong, and southern coastal basins (Fig. 1a). The first big reservoirs (mainly, Srinagarind - 18 $km^3$, Kenyi - 13.6 $km^3$, Bhumibol - 13.5 $km^3$, Sirikit - 9.5 $km^3$, Khao Laem - 8.8 $km^3$, and Nam Ngum - 7.0 $km^3$) were built between 1964 and 1985, increasing the aggregated storage capacity from 0 to ~75 $km^3$ in about twenty years (Fig. 1c). During the following 15 years (1986-2000), mostly small reservoirs were constructed, except for Rajjaprabha (5.6 $km^3$), which started to operate in 1987. Until 2000, the cumulative storage from 60 reservoirs in Mainland Southeast Asia was thus ~85 $km^3$. The construction of 125 new reservoirs in the post-2000 period sharply increased the aggregated water storage by more than two-fold, reaching a storage capacity of ~180 $km^3$ at the end of 2023 (Fig. 1c). During this time, a few mega reservoirs were built, such as Xiaowan (~15 $km^3$) and Nuozhadu (~22 $km^3$) in the Upper Mekong basin, contributing significantly to the aggregated storage capacity of Mainland Southeast Asia. At present, the largest number of reservoirs is in the Lower Mekong River basin (54), followed by the Irrawaddy (29), Red River (21), Upper Mekong (20), Chao Phraya (7), and Salween (3) [Fig. 1b, d]. 51 reservoirs are located in the remaining river basins (Indicated as "Others" in Fig. 1b, d). Although, based on the design specifications, we know how much water the reservoirs can hold, when full, we need a database containing time series of reservoir storage to better support hydrological studies and water resources management. Our MSEA-Res fills this gap.

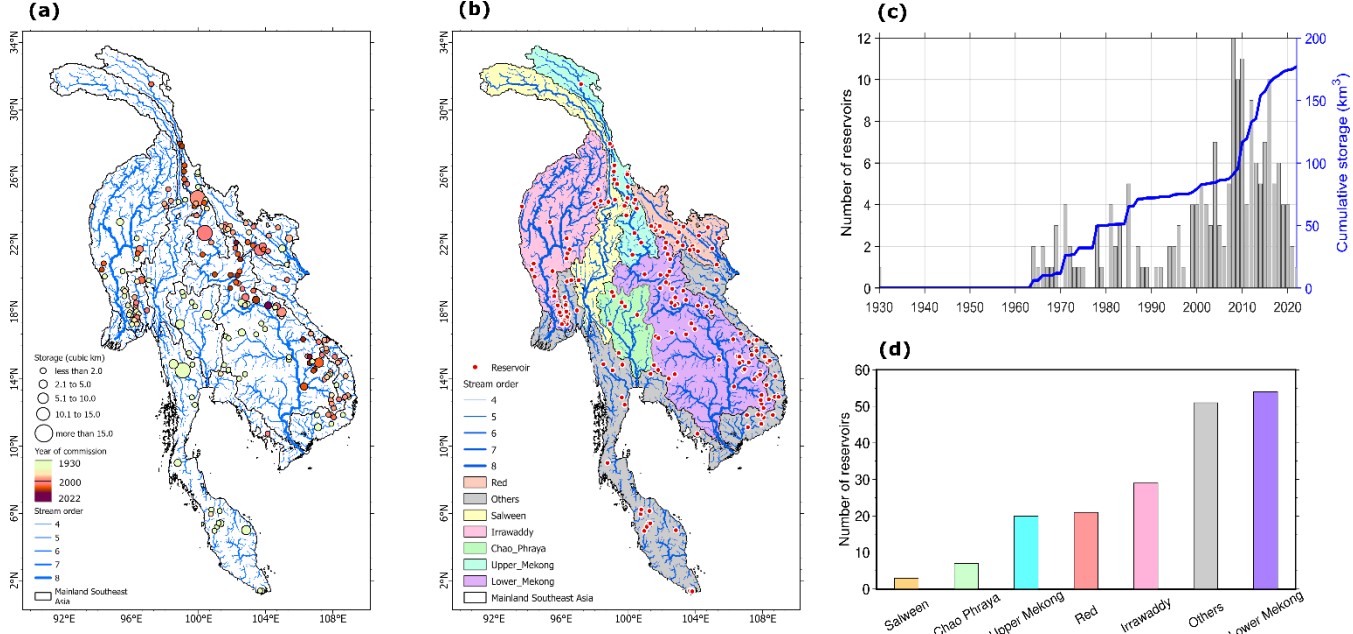

Figure 1: Spatial distribution and evolution of reservoirs in Mainland Southeast Asia. (a) Map showing reservoir storage volume (km³), where the size of the circle is proportional to the reservoir capacity while the colour represents the year of commission of the reservoirs. (b) Basin-wise distribution of reservoir location (red dots), stream network in the respective catchments, and stream order. (c) Number of reservoirs built per year and their corresponding cumulative storage capacity. (d) Basin-wise total number of reservoirs built until 2023.

## 3 Methodological framework

The procedure adopted to produce the MSEA-Res database is illustrated in Fig. 2 and can be divided into three main steps. For each reservoir, we first derive the Area-Elevation-Storage relationship (i.e., A-E curve, E-S curve, and A-S curve), then we calculate the time series of water surface area, and finally, we derive the absolute reservoir storage by combining information on the reservoir surface area (or water level, if available) with the hypsometric curves. Although water levels from satellite altimetry observations can also be used to estimate the storage volume (Zhang et al., 2014), they are only available for a few reservoirs, and they are neither consistent nor continuous in time, thus creating missing data issues (Birkett et al., 2011; Busker et al., 2019; Schwatke et al., 2015). Therefore, we worked with satellite-based water surface area, which can be produced at higher frequency (e.g., at 10-day intervals) from the Earth Observation Satellites, such as Landsat-5, Landsat-7, Landsat-8, Landsat-9, and Sentinel-2, to retrieve the reservoir's area time series. Despite Landsat having a 16-day revisit time, we could achieve a 10-day interval data because more than one Landsat mission has been active in the time domain (except for the pre-1999 period). For instance, 2013 has active sensors from the Landsat-7 ETM+ and Landsat-8 series of satellites, making it possible to achieve image composite at an interval of 10 days. Please note that there could be some months without any satellite data, resulting in storage unavailability in those months, which we filled by interpolation.

In the following sub-sections, we discuss each step-in detail, namely acquiring the raw satellite data, obtaining the hypsometric curves for different reservoirs, estimating water surface area, improving the area estimates, and finally inferring the storage time series. All steps are implemented in a Python package called *InfeRes* (publicly available on GitHub, https://github.com/Critical-Infrastructure-Systems-Lab/InfeRes/).

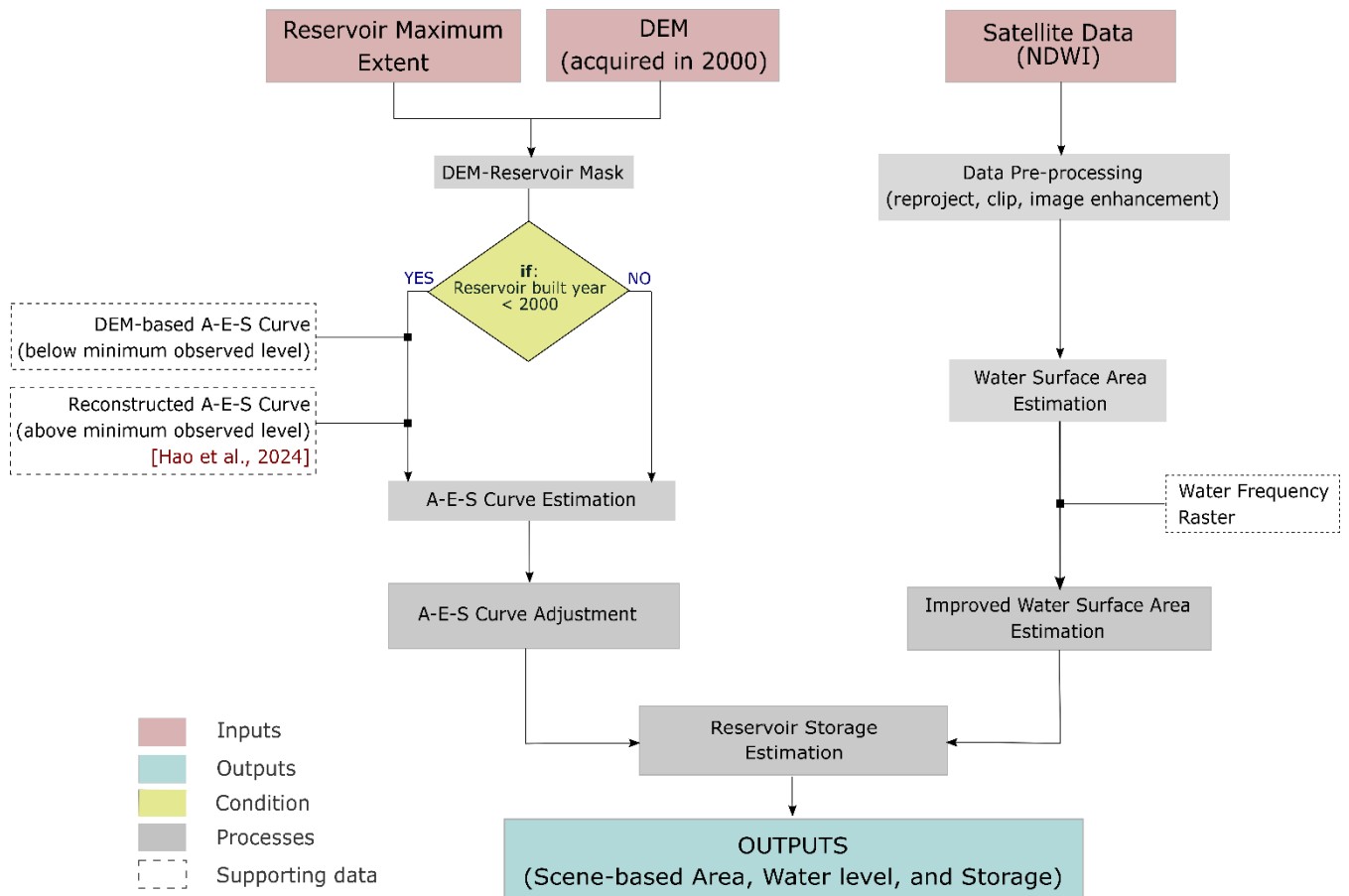

**Figure 2: Flowchart showing the methodological framework and steps taken to estimate reservoir storage from series of satellite images during 1985-2023 period. DEM is the 30m digital elevation model from the SRTM, acquired in February 2000. Normalized Difference Water Index (NDWI) is the normalized ratio between reflectance in Green and NIR bands, given by (Green-**
**NIR)/(Green+NIR), which is generally used to classify water and non-water pixels. Please note that the maximum water extent, frequency map, and NDWI images are the derived data, whereas DEM is acquired using Google Earth Engine (GEE) Python API.**

## 3.1 Acquiring input data

The process starts by obtaining the input datasets, mainly the Digital Elevation Model (DEM), Normalized Difference Water Index (NDWI) images, Water Frequency raster (FREQ), and Maximum Water Extent raster (EXT). We used the Google Earth
Engine (GEE) coding platform to derive the necessary input dataset. Please note that the maximum water extent, frequency map, and NDWI images are the derived data, whereas DEM is acquired using Google Earth Engine (GEE) Python API. For each reservoir, rectangular bounding boxes are used to fix the dimension of the dataset in the GEE. As for the digital elevation model, we used the Shuttle Radar Topography Mission (SRTM, Farr et al., 2007) DEM-Version3, an international research effort that obtained digital elevation models on a near-global scale. NASA JPL provides the SRTM V3 product at a resolution
of 1 arc-second (~30m). Unlike the DEM, the other maps (NDWI, FREQ, and EXT) were estimated from the Landsat-5 TM, Landsat-7 ETM+, Landsat-8 OLI/TIRS, Landsat-9 OLI-2/TIRS-2, and Sentilel-2 images (see Table 2 for details).

**Table 2. List of input satellite data and their specifications.**

| Category | Database | Availability | Resolution | Earth Engine Snippet |
|---|---|---|---|---|
| DEM | SRTM DEM V3 | 2000 | 30m | ee.Image("USGS/SRTMGL1_003") |
| Landsat | Landsat-5 TM | 1984-2012 | 30m | USGS Landsat 5 Level 2, Collection 2, Tier 1 ee.ImageCollection("LANDSAT/LT05/C02/T1_L2") |
| | Landsat-7 ETM+ | 1999-Present | 30m | USGS Landsat 7 Level 2, Collection 2, Tier 1 ee.ImageCollection("LANDSAT/LE07/C02/T1_L2") |
| | Landsat-8 OLI/TIRS | 2013-Present | 30m | USGS Landsat 8 Level 2, Collection 2, Tier 1 ee.ImageCollection("LANDSAT/LC08/C02/T1_L2") |
| | Landsat-9 OLI-2/TIRS-2 | 2021-Present | 30m | USGS Landsat 9 Level 2, Collection 2, Tier 1 ee.ImageCollection("LANDSAT/LC09/C02/T1_L2") |
| Sentinel | Sentinel-2 | 2016-Present | 30m (resampled) | Harmonized Sentinel-2 MSI: MultiSpectral Instrument, Level-1C ee.ImageCollection("COPERNICUS/S2_HARMONIZED") |

The Green (G) and Near-infrared (NIR) bands from the satellite sensors (Landsat and Sentinel) are used to calculate NDWI [i.e. (G-NIR)/(G+NIR)] – as proposed by McFeeters, (1996) – for the available scenes, collectively covering the study period 1985-2023. Shorter wavelength bands, Green (G) and Near-Infrared (NIR) can be affected by the presence of clouds – especially on rainy days – and so, NDWI. Therefore, getting a complete view of reservoir extent from a cloud-affected NDWI image becomes significantly challenging (Hou et al., 2024; Vu et al., 2022). To address this issue, we first filtered the Earth
Engine Image Collection based on cloud threshold (Band Quality, BQ) and selected only those images that have less than 80% cloud coverage. We also made NDWI composites from available Landsat (1985-2023) and Sentinel (2016-2023) images at 10-day intervals, which is the average of NDWI images in a given time interval (10 days in our case). For example, if we have three NDWI images with a grid cell having values of 0, 1, and 0, then the NDWI value in the composite image is 0.33. Please

note that there can be a maximum of three composite images in each month (i.e., only from Landsat) during the period 1985-2015. On the other hand, it can have a maximum of six images per month (three from Landsat and three from Sentinel) in 2016-2023. Making a composite of NDWI images maximizes the chances of getting more cloud-free pixels than individual NDWI images.

To obtain the water frequency (FREQ) and maximum water extent (EXT) raster maps, we first create the binary NDWI images available between 2013 and 2023 from the Landsat and Sentinel image collection in the GEE environment. Positive NDWI values are considered an approximation for water pixels (with a value of 1), while negative NDWI values are non-water pixels (with a value of 0). More specifically, we use a threshold slightly above zero (e.g., 0.1) to classify water and non-water pixels in the NDWI image. In general, a positive value (>0) indicates a water pixel, and using a higher threshold (e.g., 0.1) increases the likelihood of identifying water pixels accurately. While some water pixels with NDWI values between 0 and 0.1 might be misclassified as non-water, this effect is negligible when creating composites. The FREQ layer is created by making a composite of all binary NDWI images (more than 200 images from the Landsat and Sentinel collections), whose cloud percentage is less than 20% (i.e., clear sky condition) and by dividing it by the total number of selected images (cloud percentage <20%). We multiply the FREQ layer by 100 to get the percentage of water present at each pixel. For example, if three NDWI images make a composite image of value 0.33 at any grid, the FREQ value for that grid cell will be 33.3%. Please note that there can be only one FREQ raster (image), which is derived by averaging all the binary NDWI images (cloud percentage <20%) available over the reservoir. Subsequently, the EXT layer is created by simply taking the largest extent of ones in all binary NDWI images available between 2013 and 2023. For example, if we have three NDWI images with a grid cell having values of 0, 1, and 0, then the EXT value will be 1 for that grid.

To make the estimates more reliable and robust, we also validated our maps with that of the Global Surface Water Dataset (GSWD) (Pekel et al., 2016), which showed an excellent agreement ($R^2 = 0.98$) between EXT and GSWD maximum extent maps across the 186 reservoirs (Fig. S1). We also compared EXT and FREQ maps spatially, for two randomly selected reservoirs i.e. Sirikit and Shringarind, which also confirmed the reliability of water frequency (FREQ) and maximum water extent (EXT) raster maps that we derived from GEE (Fig. S2 and S3). Overall, we assemble three raster layers (DEM, FREQ, and EXT) and scene-based NDWI images for each of the 186 reservoirs, which we process further to estimate the absolute reservoir storage time series.

### 3.2 Area-Elevation-Storage curves

Deriving the relationship between the area, elevation (or water level), and storage (A-E-S relationship) of a reservoir is crucial. This step relies on the bathymetry information, which further depends on the time of acquisition of the DEM. Considering that the SRTM-DEM was acquired in February 2000, reservoirs built after 2000 have complete bathymetry information; thus, the A-E-S relationship after the year 2000 can easily be derived. Since the majority of the reservoirs (~70%) in Mainland Southeast

Asia were built after 2000, we obtained the Area-Elevation-Storage (A-E-S) curves from the DEM. For each reservoir, the elevation range for the A-E-S curves was defined by the minimum and maximum DEM values within the reservoir's extent. The area at each elevation level was determined by contouring, while the corresponding absolute storage was estimated by cumulatively summing the areas across the elevation range. For the remaining 30% of the reservoirs built before 2000, the DEM cannot be applied directly to estimate E-A-S curves. This is a common problem in the existing studies for estimating absolute storage for reservoirs built before 2000 (Busker et al., 2019; Gao et al., 2012; Hou et al., 2024; Khazaei et al., 2022; Yigzaw et al., 2018). Although previous studies have used various modeling approaches based on simplified geometric assumptions to overcome this limitation (Fang et al., 2023; Hou et al., 2024; Khazaei et al., 2022; Yigzaw et al., 2018), results often do not meet the level of accuracy required for basin-scale water management modeling and decision-making. To address this problem, we banked on a recently-released database of global reservoir area-storage-depth derived through deep learning-based bathymetry reconstruction (GRDL; Hao et al., (2024)), which provides reliable bathymetry information for the 7,250 GRanD reservoirs across the globe. We thus utilized the GRDL database to obtain A-E-S curves for the remaining 60 reservoirs (Fig 2).

## 3.3 Water surface area estimation

We used the Landsat and Sentinel-based NDWI images downloaded from the Google Earth Engine platform (see section 3.1 for details) to estimate the reservoir water surface area. A locally-adjusted Contrast Limited Adaptive Histogram Equalization (CLAHE) was applied to enhance the NDWI images before classification. CLAHE (Reza, 2004) is a variant of Adaptive histogram equalization (AHE), which takes care of over-amplification of the contrast in an image. CLAHE operates on small regions in the image (8 x 8 pixels window in our case) rather than the entire image. The size of its operational window (8 x 8) is based on the literature (Asghar et al., 2023), which suggests that CLAHE enhances the contrast and texture features of water, thereby improving the visualization of satellite images. This enhancement facilitates the classification of water and non-water pixels. We then applied the *k*-means clustering-based algorithm to classify the water pixels. We assigned a number of clusters (*k*) equal to three to classify each NDWI image to represent three different classes, i.e., water, non-water, and no data. Because of the presence of clouds and other disturbances, using the same NDWI threshold (equal to 0) in all satellite images may lead to overestimation or underestimation errors of the water surface area (Vu et al., 2022). Thus, to find NDWI thresholds for each satellite image, we resort to *k*-means clustering. Eventually, the preliminary water pixels were identified by selecting the cluster corresponding to the maximum centroid value of NDWI. The water surface area estimated from the preliminary water pixels is referred to as 'Before_area' for any given reservoir (Table 3).

We further improved the water surface area estimates by filling the cloud-contaminated pixels, which were assigned a "No Data" value in the previous steps. To this purpose, we used the algorithm for water surface area estimation developed by Vu et al., (2022), which was initially introduced by Gao et al., (2012) and Zhang et al., (2014) to extract water surface area. The algorithm uses a water frequency raster (FREQ) to fill the cloud-affected pixels over the reservoir area. We add the clear water

pixels (*k*-means clustering) and cloud-filled water pixels (Vu et al., 2022) to get a complete picture of the reservoir water surface area for each NDWI image, called 'After_area' (Table 3). Finally, we adjust the boundary water pixels of the complete reservoir water surface area, represented as 'Final_area'. Notably, if no adjustments are detected by the algorithm, the 'Final_area' remains equal to the 'After_area' (Table 3).

### 3.4 Absolute storage estimation and post-processing

Once estimated the water surface area, we subsequently used the hypsometric curves to derive the corresponding absolute reservoir storage volume. Based on different processing stages, we post-processed the storage time series into three levels (Level-0, Level-1, and Level-2). For each reservoir, Level-0 corresponds to the scene-based (instantaneous) raw outputs of absolute reservoir storage, which have been derived from the available satellite images. We then performed a simple box plot analysis on Level-0 data to remove the outliers, creating the so-called Level-1 data. Level-0 data are provided to give users

the flexibility to generate their own Level-1 data using alternative outlier removal algorithms, if needed. Note that in our case, Level-1 data are created using a generalized box-plot framework for quality control that is not specifically designed for each reservoir; therefore, on a case-to-case basis, some values in the storage time series may still be considered outliers—they can be removed manually or with the aid of other data analysis algorithms. Therefore, the improvement in Level-1 data compared to Level-0 data varies between the reservoirs. To quantify it, we calculated the $R^2$ and nRMSE for level-0 and level-1 data of

the 20 reservoirs for which we have the observed storage. The detailed analysis of the 20 selected reservoirs is presented below in section 4.4 (Fig. 7 and Table S2). We found that the nRMSE decreased and $R^2$ increased from Level-0 to Level-1, suggesting that the outlier removal process can further enhance the quality of the data (Fig. S4). Considering the demand for ready-to-use data for several applications (e.g., hydrological modelling), we further processed the Level-1 data to create continuous daily time series of absolute reservoir storage (called Level-2 data) using a non-linear (i.e. spline) interpolation technique, followed

by data smoothening (moving mean method). It is important to note that the interpolation technique incorporates all available data points, including a few outliers, which introduces a higher level of uncertainty in the Level-2 data. Despite this, we undertook validation of the storage time series to strengthen confidence in our estimations.

### 3.5 Validation of reservoir storage

    We adopted two validation approaches. The first approach is direct validation, where we compare and validate our estimated

storage volume against observed reservoir storage. The second is indirect validation, where we use altimetry-converted storage to validate our time series of reservoir storage. Acquiring observed reservoir storage is challenging in MSEA because of the institutional and organizational data-sharing policies and restrictions, leading to a poor network of public data repositories for reservoir data. The only exception is the Thailand National Hydroinformatics Data Centre, which releases daily reservoir storage information to the public domain (National Water Database (NWD) - https://www.thaiwater.net/). We took the

opportunity to download observed storage data from the NWD portal for 20 reservoirs in Thailand, and then compared these data with our storage estimates.

For indirect validation, we used reservoir water level data measured by satellite-based altimeters such as TOPEX/Poseidon; Jason-1, Jason-2, and Jason-3; ENVISAT; ERS-1 and ERS-2; and Sentinel-3 and Sentinel-6, which have proven useful in measuring water levels in lakes and reservoirs (Birkett, 1998; Frappart et al., 2006; Santos da Silva et al., 2010). Specifically, we acquired the compiled time series of radar-altimetry-derived surface water elevation from the Database for Hydrological Time Series of Inland Waters (DAHITI- https://dahiti.dgfi.tum.de/) (Schwatke et al., 2015), and the Global Reservoirs and Lakes Monitor (GREALM- https://ipad.fas.usda.gov/cropexplorer/global_reservoir/) (Birkett et al., 2011). We took 20 reservoirs across Mainland Southeast Asia – for which altimetry observations are available – to indirectly validate our estimated storage time series. Before carrying out the comparison, the altimetry-derived surface water levels were first converted to their corresponding storage time series based on the Elevation-Storage relationship.

## 4 Results

### 4.1 Structure of the MSEA-Res database

The reservoir's information in the database is divided into static and dynamic components (Fig. 3 and Table 3). For each reservoir, static information is further divided into four categories: i) Area-Elevation-Storage relationship (hypsometric curves), (ii) reservoir extent, iii) water frequency (mean inundation frequency for each pixel), and iv) reservoir's characteristics such as location (longitude and latitude), year of commission, area ($km^2$), water level (m), and storage (million $m^3$). Note that for area, level, and storage, static information includes minimum, mean, and maximum. On the other hand, dynamic information consists primarily of the sub-monthly time series of absolute reservoir storage. We did not separately provide the water-level and surface area time series, as they can easily be derived from the Area-Elevation-Storage curve for any given storage volume. In the subsequent sections, we use Level-1 and Level-2 data to analyze and validate the storage time series. Note that, for each reservoir, the data are processed within the period 1985-2023. If the year of commission of a reservoir is 2015, then the storage time series is estimated between the years 2010 and 2023, assuming a maximum of five years as the filling period. All storage time series and other related information are publicly available in the MSEA-Res database at https://zenodo.org/records/14844580 (Mahto et al., 2025).

## Mainland Southeast Asia's water reservoirs (MSEA-Res) catalogue

**Static components**

- **Area-Level-Storage relationship** (curves)
- **Reservoir extent** (georeferenced raster image)
- **Water frequency** (georeferenced raster image)
- **Reservoir's characteristics** (location, year of commission, area, water level, and absolute storage

**Dynamic components (Storage time-series)**

- **Level-0:** Scene-based raw outputs
- **Level-1:** Scene-based filtered outputs (outliers removed)
- **Level-2:** Continuous daily outputs (interpolated and smoothened)

**Figure 3: Catalogue of the MSEA-Res database. Please note that the dynamic components (storage time series) are available from five years before the year of commission.**

**Table 3. Reservoir attributes in the MSEA-Res database.**

| Category | Datatype | Attributes | Description |
|---|---|---|---|
| Static components | Area-Elevation-Storage relationship | Level_m | Water level |
| | | Area_sq_km | Water surface area |
| | | Storage_cubic_km | Absolute storage |
| | Reservoir extent | | Georeferenced image (.TIFF) |
| | Water frequency | | Georeferenced image (.TIFF) |
| | Reservoir's characteristics | Sl_No | Serial number as per MSEA-Res database |
| | | GRAND_ID | Identification number in the GRanD database (Lehner et al., 2011). For a non-GRandD reservoir, the value is 9999. |
| | | Longitude | Longitude in degrees decimal |
| | | Latitude | Latitude in degrees decimal |
| | | Year_of_commission | Year of commission of the reservoir |
| | | Area_min_sqkm | Minimum water surface area ($km^2$) |
| | | Area_avg_sqkm | Average water surface area ($km^2$) |
| | | Area_max_sqkm | Maximum water surface area ($km^2$) |
| | | WL_min_m | Minimum surface water level (m) |
| | | WL_min_m | Average surface water level (m) |

| | | | |
|---|---|---|---|
| | | WL_min_m | Maximum surface water level (m) |
| | | Storage_min_cubic_km | Minimum water storage (km$^3$) |
| | | Storage_min_cubic_km | Average water storage in (km$^3$) |
| | | Storage_min_cubic_km | Maximum water storage in (km$^3$) |
| Dynamic components (Storage time series) | Level-0 | ID | Satellite data identification number (L0= Landsat and S2= Sentinel) |
| | | Date | Image collection data |
| | | Cloud_percentage | Percentage of cloud cover over the reservoir |
| | | Quality | Quality control indicator (1= Good, 0= Bad) |
| | | Before_area | Instantaneous water surface area before improvement (km$^2$) |
| | | After_area | Instantaneous water surface area after improvement (km$^2$) |
| | | Final_area | Instantaneous water surface area after final check (km$^2$) |
| | | dem_value_m | Instantaneous surface water level (m) |
| | | Tot_res_volume_km3 | Instantaneous water storage after a final check (km$^3$) |
| | Level-1 | Same as Level-0 | Same as Level-0 |
| | Level-2 | Date | Daily dates |
| | | Storage_km3 | Interpolated instantaneous water storage (km$^3$) |

## 4.2 Hypsometric curves and storage time series

In this section, we illustrate one of the static components of the MSEA-Res database, i.e., the Area-Elevation-Storage relationship (see Table 3 for details), where elevation corresponds to the reservoir's water level relative to mean sea level in meters (m a.s.l). In our database, we provide the hypsometric curves for each of the 186 reservoirs. Here, we further illustrate seven curves (Fig. 4); one reservoir for each major river basin. The seven selected reservoirs (basin) are Longjiang (Irrawaddy), Nuozhadu (Upper Mekong), Son La (Red), Mobye (Salween), Sirikit (Chao Phraya), Sringarind (Other basins), and Xe Kaman1 (Lower Mekong). The Area-Elevation (A-E curve) and Storage-Elevation (S-E curve) relationships are shown (Fig. 4). These hypsometric curves represent the variability in reservoir's storage and area, which results primarily from the diverse topography characterising the basins and reservoir locations.

For the same seven reservoirs, we then illustrate the dynamic components of the MSEA-Res database – time series of reservoir storage at different processing levels, i.e., Level-0 (raw outputs), Level-1 (removal of outliers from Level-0), and Level-2 (smooth interpolation of Level-1) (Fig. 5). The storage time series data can be used to infer meaningful information on the storage dynamics, including filling patterns, fluctuations, and response to wet and dry years. Looking at the filling patterns, for instance, Xe Kaman1 (2016) took almost four years to store more than 3 km$^3$ of water and reach its normal operating

conditions (Fig. 5g). The Longjing (2010) reservoir was filled in roughly one year (Fig. 5a). By combining this information with inflow data, one could easily estimate the impact of reservoir filling strategies on downstream water availability—a rather contentious matter in transboundary river basins (Vu et al., 2022; Wheeler et al., 2016; Zaniolo et al., 2021). The time series

also reveal the 'typical' behaviour of reservoir storage in Southeast Asia, with seasonal fluctuations between minimum and maximum operating levels driven by the drastic changes in the intra-annual water availability characterizing this region (i.e., a wet season between June and November followed by a drier period between December and May) (Nguyen et al., 2020). Importantly, the time series also reveal inter-annual changes in water storage, which are largely caused by hydrological-regime variability – a point further discussed in Section 4.5.


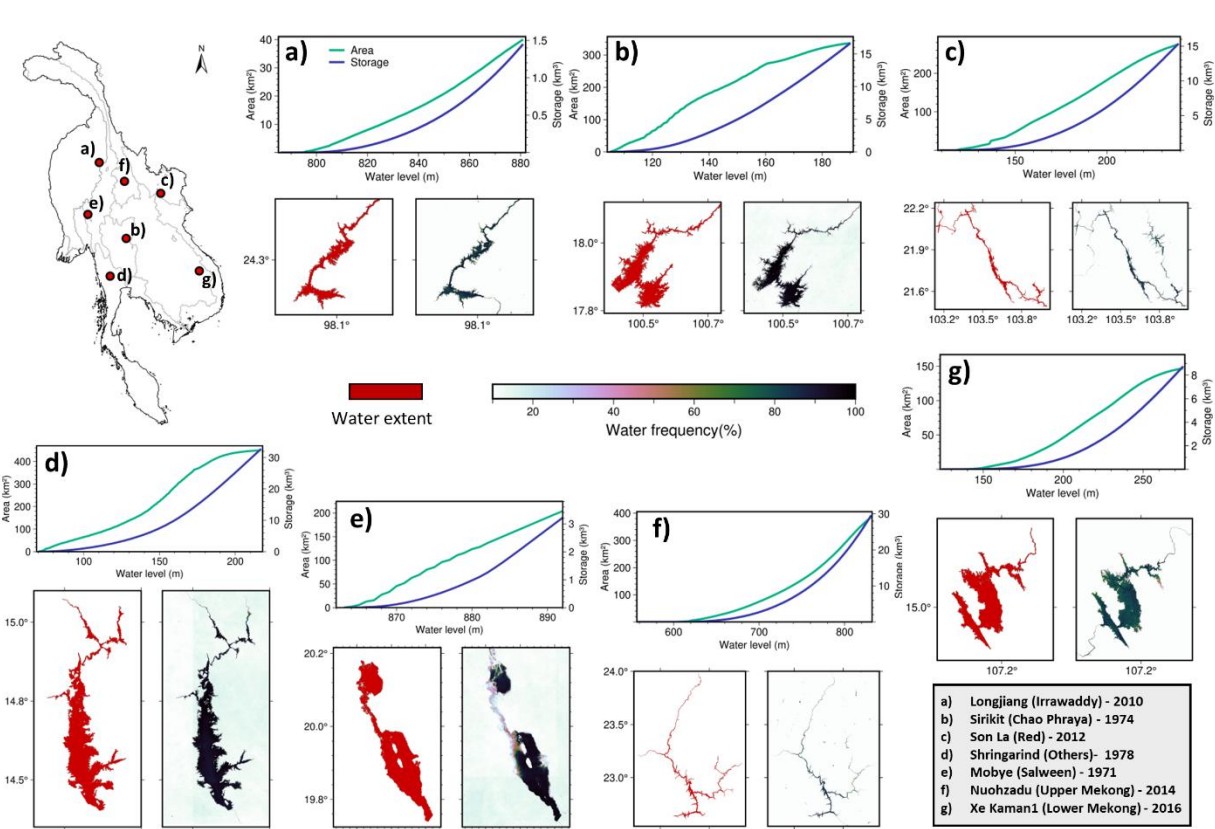

**Figure 4: Illustration of the static components of MSEA-Res database (Area-Elevation-Storage relationship) for seven reservoirs, one in each of the major river basins. In each panel, Elevation-Area (E-A) and Elevation-Storage (E-S) curves are shown in green**

**and blue, respectively. The dates refer to year of commission of the reservoirs.**

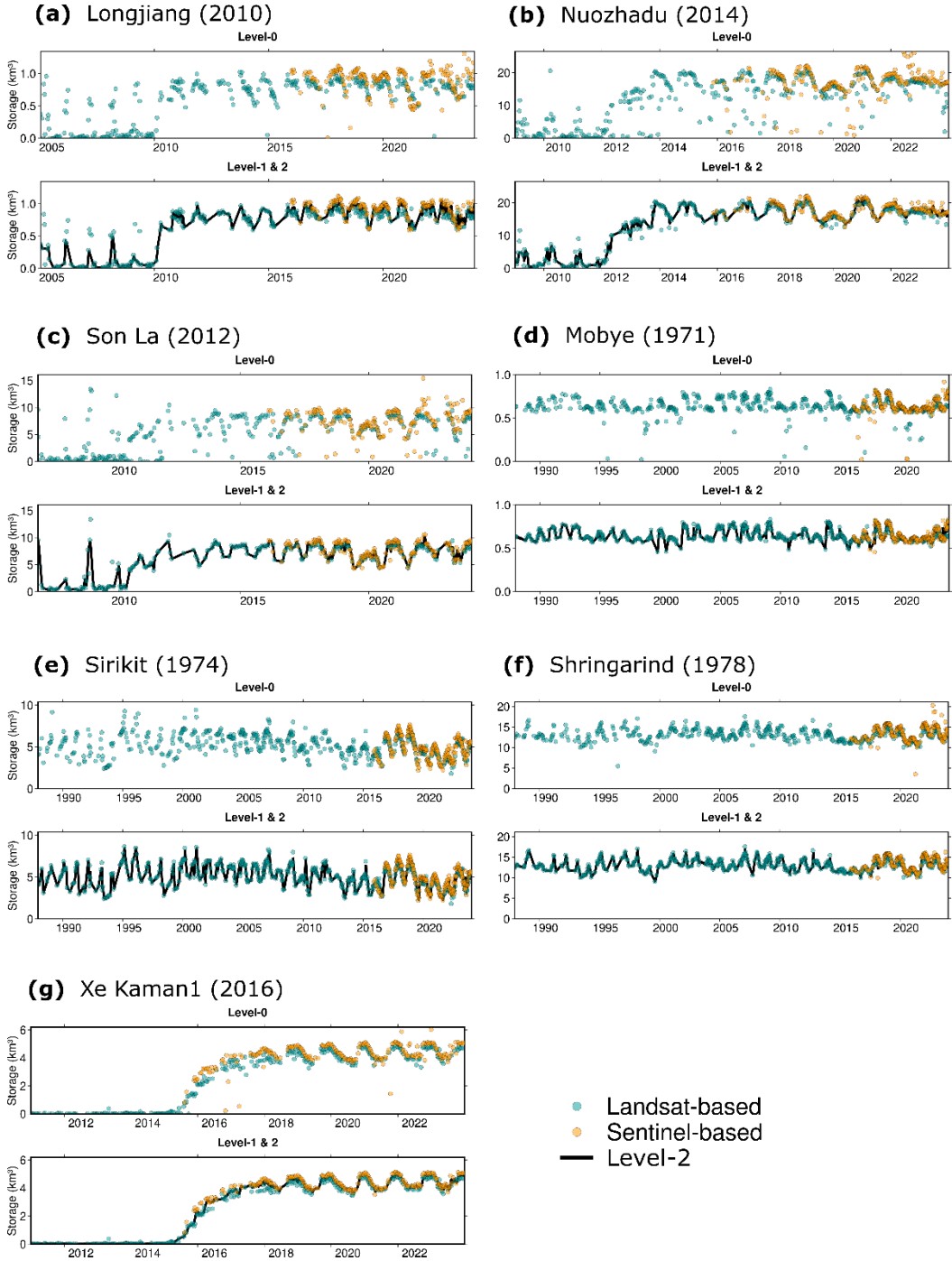

**Figure 5: Illustration of reservoir storage time series – i.e., the dynamic components of the MSEA-Res database, for the seven**
**selected reservoirs. Each panel (a-g) corresponds to a reservoir. For each panel, we report the scene-based reservoir storage (km³)**

time series at Level-0, and the storage time series at Level-1 (after removing the outliers) overlapped with Level-2 (after interpolation and smoothening).

### 4.3 Basin-wise reservoir storage analysis

We used all Level-1 data to analyse the basin-wise evolution and dynamics of reservoir storage in Mainland Southeast Asia. Specifically, we calculated the total volume of water ($km^3$) stored in all reservoirs for each of the main seven river basins, namely Irrawaddy, Upper Mekong, Red, Salween, Chao Phraya, Lower Mekong and "other basins" lumped together (Fig. 6). We found that the aggregated storage of all reservoirs in the Upper Mekong basin has increased by more than eight times (800% increase) in just five years (between 2010 and 2015) (Fig. 6b). Nuozhadu (22 $km^3$) and Xiaowan (15 $km^3$) are the main

contributors to such increase, as they account approximately for 95% of the basin-total storage, whereas the remaining 18 reservoirs contribute just 5% (Fig. 6b). Since the construction of Nuozhadu and Xiaowan, more reservoirs have been built in the Upper Mekong; yet, their capacity is smaller than the one of these two mega reservoirs (e.g., Miaowei, 0.66 $km^3$). A seasonal fluctuation of storage is common across all basins, as the monsoon season has a similar precipitation pattern across the MSEA region (Ha et al., 2023; Skliris et al., 2022).


Results further illustrate the spatio-temporal variability in reservoir construction across Mainland Southeast Asia. After 2017, all basins – except for the Lower Mekong – reached a plateau, with no significantly increasing trends in their aggregated reservoir storage (Fig. 6). For instance, 1998-2015 was the period in which a series of reservoirs were constructed in the Irrawaddy basin, increasing the aggregated storage volume form ~2 $km^3$ to 10 $km^3$ (500% increase). Similarly, it was in 2005-

2015 and 2010-2017, respectively, that the largest reservoirs were built in Red river (300% increase) and in the other-coastal basins (35% increase) (Fig. 6c, f). The aggregated reservoir storage in the Lower Mekong basin has instead increased since 2009 (Fig. 6g). Two river basins – Salwaeen and Chao Phraya – show no significant change in the aggregated reservoir storage in last four decades (Fig. 6d, e). In fact, the storage volume in Chao Phraya has been found to be substantially reduced by ~15% in the post-2010 (Fig 6e), due to persisting drought conditions during which both Bhumibol and Sirikit reservoirs showed

a continuous decline in storage (Fig. S6b, Fig 5e). Putting all 186 reservoirs all together, we find that the aggregated average reservoir storage in Mainland Southeast Asia has increased significantly, from 70 to 160 $km^3$ (130% increase), during the period 2008-2017. Presently, it is approximately 175 $km^3$ (Fig. 6h). Additional details regarding the temporal evolution of reservoir storage in MSEA are reported in Fig. S5.

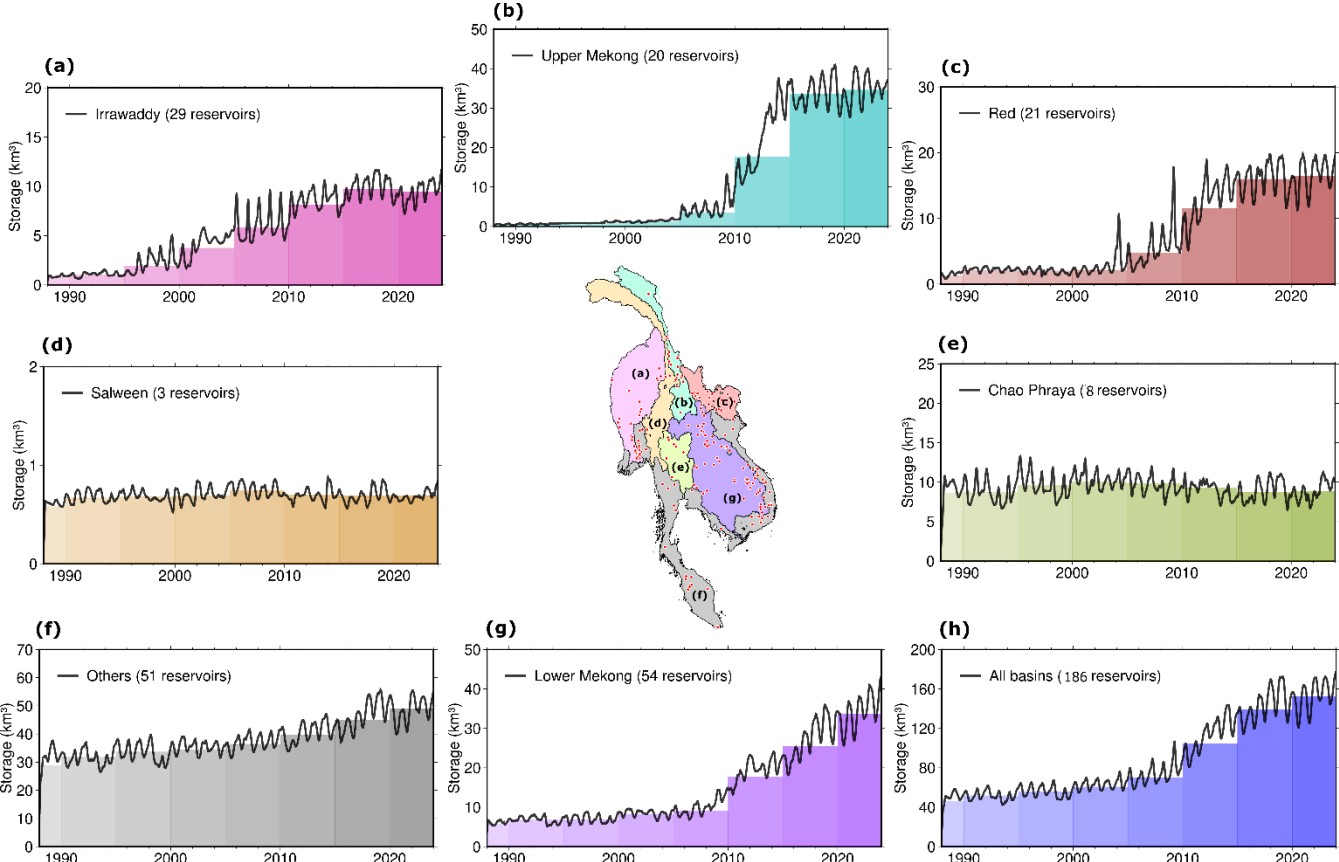

**Figure 6: (a-g) Aggregated storage time series in Irrawaddy, Upper Mekong, Red, Salween, Chao Phraya, Lower Mekong and other minor river basins, during the 1985-2023 period. (h) Aggregated storage time series of all the 186 reservoirs in Mainland Southeast Asia. The color gradient in each panel represents the average storage with a 5-year interval. Please note that the aggregated storage is the total volume of water (km³) stored in all reservoirs at a given time in each river basin.**

## 4.4 Validation

We validated the generated storage time series – Level-1 data – with the observed reservoir storage (direct validation) and altimetry-converted storage (indirect validation). As explained in Section 3.4, we first collected observed storage for 20 reservoirs in Thailand from the National Water Database, the only publicly available storage database in the MSEA region. We then compared the estimated and observed storage based on two metrics – coefficient of determination ($R^2$) and normalized (by reservoir's total storage) root-mean-square-error (nRMSE).

Despite the lack of actual bathymetry for most reservoirs in Thailand (since they were built before 2000), we found a good agreement between estimated and directly observed storage in most reservoirs (Fig. 7a, b). Sirikit and Shringarind showed

very good agreement, with $R^2 > 0.8$ and nRMSE < 9.5% for both reservoirs (Fig. 7c, d). Notably, 10 out of 20 reservoirs show an $R^2$ greater than 0.7 (average $R^2 = 0.77$ and average nRMSE = 14.2%) (Fig. 7a, b; Table S1). Excluding three reservoirs with lower performance (Bang Lang, Rajjaprabha, and Bhumibol), the average $R^2$ and nRMSE of the remaining 17 reservoirs is 0.68 and 17%, respectively (Table S1), suggesting that the framework works well for reservoirs characterized by varying A-E-S curves and sizes. For instance, the validation shows a strong agreement for both Khao Laem (~8 km$^3$) and Lamphraphloeng

(~0.1 km$^3$), with $R^2 > 0.77$ and nRMSE < 18% for both reservoirs (Fig. S6, Table S1). As expected, the average $R^2$ and nRMSE across all 20 reservoirs are approximately 0.6 and 18.6%, respectively (Table S1).

To make the evaluation more robust, we indirectly validated our storage time series using altimeter observations collected from the DAHITI and G-REALM databases. The water level time series acquired from various altimeters was converted to

the corresponding storage-time series using the Elevation-Storage (E-S) curve (see Section 3.4). We collected water level observations for 20 reservoirs across the MSEA region for which the altimetry passes were available for at least five years. The comparison between time series shows that 14 of 20 reservoirs have an $R^2$ larger than 0.7 (average $R^2 = 0.80$ and average nRMSE = 11.7%), suggesting a good match between estimated and altimetry-converted storage time series (Fig. 8a, b). The average $R^2$ and nRMSE are 0.63 and 13.3%, respectively, when considering all 20 reservoirs together (Fig. S7, Table S2). The

storage time series comparison for two of the largest reservoirs [Sirikit ($R^2 = 0.70$, nRMSE = 17%)] and Nuozhadu [($R^2 = 0.96$, nRMSE = 6.4%)] are shown in Fig. 8c and Fig. 8d, respectively.

The underperformance of certain reservoirs can likely be attributed to two key factors. First, potential inaccuracies in the hypsometric curves may introduce errors when converting inferred water surface area into absolute reservoir storage. Second,

the quality of satellite-derived NDWI data, particularly cloud-free image availability and gap filling, can significantly impact accuracy. Enhancing satellite image pre-processing through techniques such as contrast stretching and histogram equalization could improve data quality and, in turn, refine reservoir storage estimations. Addressing these challenges will be crucial in further optimizing the framework's reliability across diverse hydrological settings. Despite these challenges, the direct and indirect validation metrics suggest that the *InfeRes*-derived storage data can be reliably used for water storage-related analysis

on a weekly to yearly time scale.

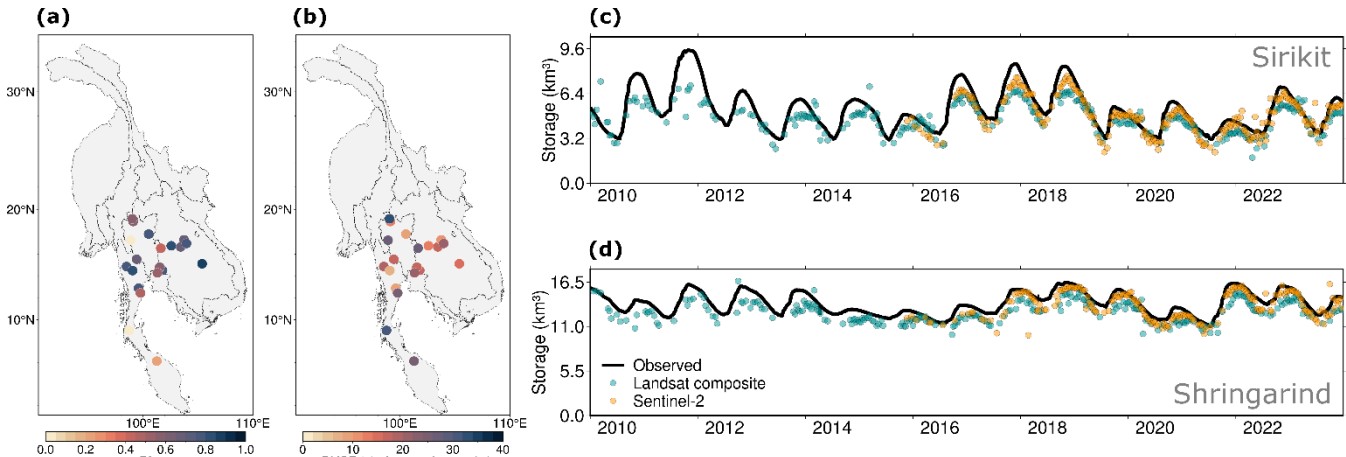

**Figure 7: Direct validation of the inferred storage time series against local observations. (a-b) Spatial distribution of the coefficient of determination (R²) and nRMSE, respectively. (c) Comparison of the absolute storage time series for Sirikit reservoir during the period 2010-2023. (d) Same as (c), but for Shringarind reservoir.**

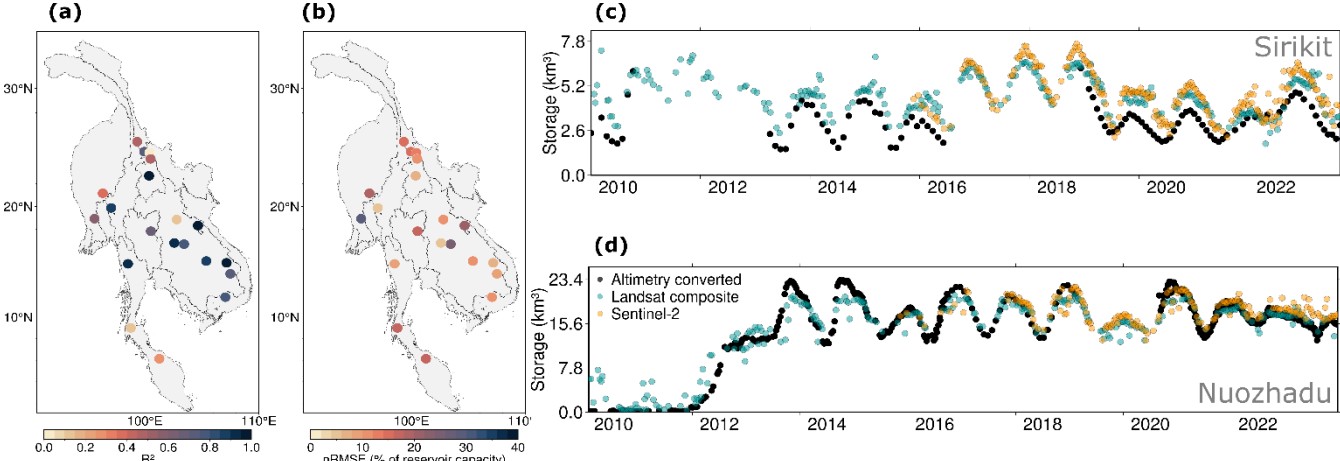

**Figure 8: Indirect validation of the inferred storage time series against the altimetry-converted storage (water level is converted to the corresponding storage using the Elevation-Storage curve). (a-b) Spatial distribution of the coefficient of determination (R²) and nRMSE, respectively. (c) Comparison of the absolute storage time series for Sirikit reservoir during the period 2010-2023. (d) Same as (c), but for Nuozhadu reservoir.**

### 4.5 Example application: 2019-2020 drought's impact on water storage

We finally used the estimated storage time series (Level-1 data) to showcase an example application of the MSEA-Res database. Studies reported that the 2019-2020 drought in the MSEA region seriously impacted agriculture, water resources, and hydropower generation (Ha et al., 2022, 2023). Banking on the new developed data, we analysed the impact of the 2019-

2020 drought on surface water storage across the region by utilizing precipitation data from the Climate Hazards group Infrared Precipitation with Stations [CHIRPS; (Funk et al., 2015)] and storage anomalies for all 186 reservoirs.


The precipitation anomalies (%) in 2019 and 2020 with respect to the reference period 1981-2023 are very pronounced (Fig. 9a and 9b). In 2019, Mainland Southeast Asia experienced wide-spread below-average precipitation conditions, with rainfall significantly lower than the historical average in most areas, with some regions facing a decrease as high as -40% (Fig. 9a). Nearly 30% of the MSEA region suffered from more than five months of drought, impacting, in particular, Cambodia and

Thailand (Fig. 9a). In contrast, 2020 showed a more mixed pattern, with several areas experiencing above-average precipitation while others continuing to have below-average levels (Fig. 9b). Overall, these severe drought conditions damaged nearly 40% of the rainfed rice area (Ha et al., 2023) and also threatened the surface water storage in lakes and reservoirs (ReliefWeb report, 2020; Ha et al., 2022).

To quantify the impact of the drought on the reservoir's storage volume, we estimated the reservoir storage anomalies in 2019 and 2020 against the reference period 2017-2023. The anomalies in storage volume of the selected reservoirs for 2019 and 2020 are mostly negative (Fig. 9c and 9d). In 2019, 120 of 186 reservoirs (65%) exhibited negative storage departures, reflecting reduced water levels consistent with the observed precipitation deficit (Fig. 9c). These storage departures ranged up to -40%, highlighting significant impacts on water availability in the region. Many lakes in Cambodia and Thailand were

indeed hardly hit by drought conditions, resulting in below-average levels. Reservoirs situated in the eastern basins (e.g., Mekong, Red River) were affected the most, compared to the reservoirs in the western part, where some reservoirs showed positive storage anomalies (Fig. 9c). Storage conditions were worsened in 2020, with 144 of 186 reservoirs (78%) exhibiting negative storage departures, primarily due to the combined effects of precipitation deficits in both 2019 and 2020 (Fig. 9d). Interestingly, we noticed some discrepancy between the spatial distribution of the precipitation and water storage anomalies

(Fig. 9), likely due to the topology of the cascading reservoir system. In other words, some reservoirs located in regions characterized by positive precipitation anomalies, but may receive limited inflow from upstream reservoirs located in regions affected by droughts. Except for reservoirs in the Upper Mekong basin, all other reservoirs experienced storage anomalies ranging between -5% to -40% (Fig. 9d). This is in line with direct observations, as the reservoirs in nine provinces of Thailand -- Chiang Mai, Uthai Thani, Chaiyaphum, Khon Kaen, Nakhon Ratchasima, Buri Ram, Suphan Buri, Lop Buri, and

Chachoengsao – were reported to reach low storage values (ReliefWeb report, 2020; Danial R, 2021). As a result, Thailand experienced the worst water crisis in the past 40 years, with 25 provinces declaring drought disaster zones (Danial R., 2021). Moreover, the 2019-2020 water shortage increased the political tensions among countries, particularly in the Upper-Mekong Region, thus exacerbating the impact of the drought in the lower basins. Overall, analyses like this one illustrate the importance of working with detailed information on reservoir operations when analysing the impact of droughts: aside from the preliminary

analysis reported here, one could, for instance combine the storage data with a hydrological model to investigate the drought impact on the Mekong level, which was reported to have reached the lowest level in almost 100 years (MRC report, 2020).

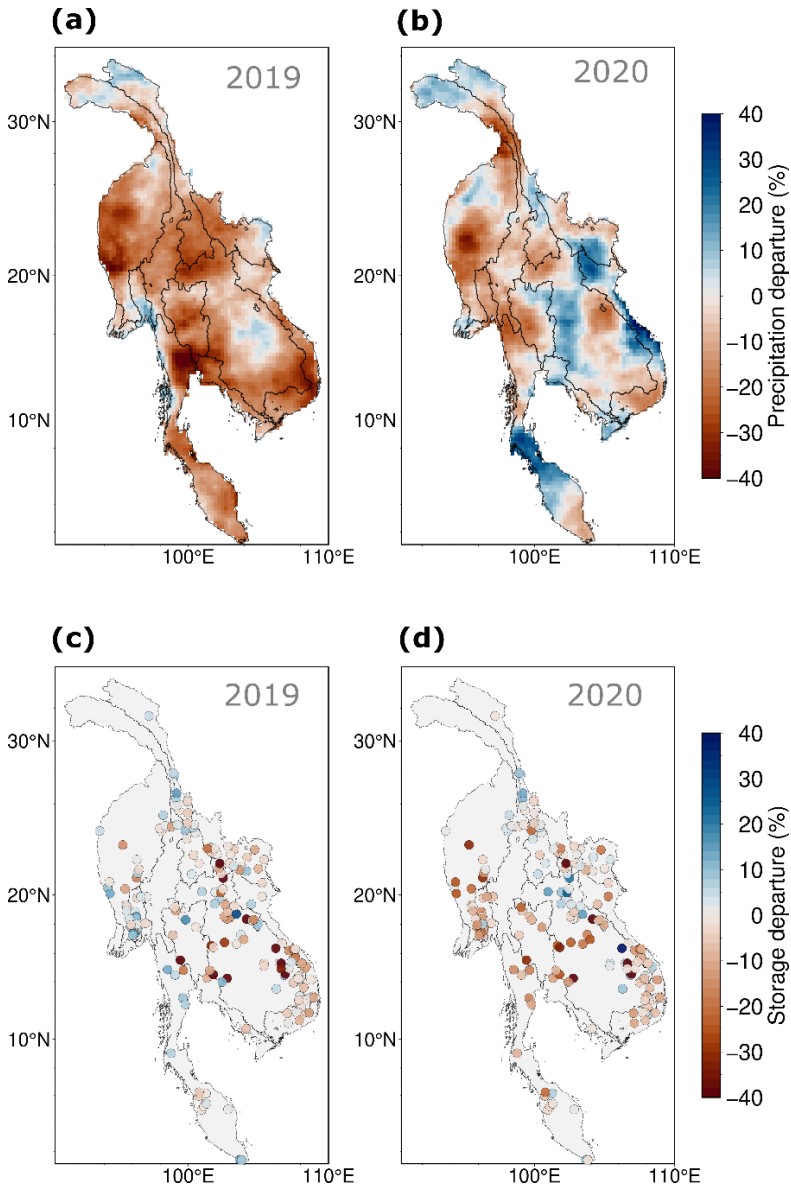

**Figure 9: Precipitation and water storage anomalies during the 2019-2020 drought in Mainland Southeast Asia. (a-b) Spatial variability in the precipitation anomalies (%) in 2019 and 2020. (c-d) Same as (a-b), but for reservoir storage anomalies. The anomalies of precipitation were estimated against the reference period 1985-2023, whereas, for storage anomalies, the reference period is 2017-2023.**

## 5 Discussion and Conclusions

We produced time series of absolute storage for 186 reservoirs (with capacity larger than 0.1 km$^3$) in Mainland Southeast Asia for the period 1985-2023 with an aggregated storage capacity of nearly 175 km³ by the year 2023, which corresponds to about 60mm of water storage over the entire mainland Southeast Asia region. The reservoir time series were reconstructed using optical remote sensing data (NDWI) from Landsat composite and Sentinel-2 with a 10-day temporal resolution. The reservoir locations and other attributes, such as design capacity, year of commission, and maximum surface area, were retrieved by combining GRanD v1.3, the Mekong Dam database (Ang et al., (2024)), and the Global Dam Tracker (GDAT) database. For each reservoir, we generated (i) scene-based NDWI raster image, (ii) water frequency raster, (iii) maximum water extent raster, and (iv) elevation raster (i.e., DEM). A Python package called '*InfeRes*' was created to automatically download and process all satellite images using the Google Earth Engine Python API. The water area from the satellite data was then translated into storage values using hypsometric curves (Area-Elevation-Storage relationship) derived from the Shuttle Radar Topography Mission (SRTM) Digital Elevation Model (DEM) and bathymetry reconstructions from the GRDL database, wherever necessary.

The reconstructed database of absolute storage time series – unlike storage change metrics – offers a detailed view of reservoir status at any given time, thus providing a comprehensive and contextualized understanding of reservoir dynamics. This approach is particularly valuable for long-term monitoring (Gao et al., 2012) and planning of water resources in the region (Galelli et al., 2022; Minocha et al., 2024). Accurate absolute storage estimates allow for detecting subtle trends and shifts in water availability that could be masked by focusing solely on changes (Hou et al., 2024; Li et al., 2023). This is particularly crucial for transboundary rivers like the Mekong, where the availability of data on reservoir operations could help alleviate the water governance issues that emerged in the past years (Danial R., 2021). Another important downstream application of MSEA-Res is hydrological modelling; integrating the estimated absolute reservoir storage data into hydrological models can offer significant advances in the understanding of human-water interactions and resource management in Mainland Southeast Asia. This integration allows for refining models that simulate water management strategies (Chang et al., 2019; Chowdhury et al., 2020; Galelli et al., 2022), and flood control (Shin et al., 2020; Wang et al., 2021).

Importantly, the developed code (available at https://zenodo.org/records/14844580) and framework are not tailored to Southeast Asia, therefore enabling their application to individual studies or other regions as well as further enrichment of this inventory with new reservoirs. The publicly available reservoir time series dataset can be used directly to assess storage trends and variability under climate change, inferring reservoir operations, agricultural water management, and hydrological model's inputs, and for comparison with previous studies. The overall outcome of our study will hopefully facilitate reservoir management and related research in hydrology, environmental science, and climate studies.

Although the extraction of water surface area using optical images from Landsat and Sentinel-2 has provided valuable insights, there remains scope for further improvements. For example, other image processing techniques can be applied to further enhance the water surface estimates. This includes band normalization, adaptive filtering, and edge enhancement filters – other than Adaptive Histogram Equalization (CLAHE). Please note that CLAHE applied to NDWI images does not specifically

correct for high turbidity, shadows, aquatic vegetation, mixed land-water pixels, or seasonal vegetation effects—this remains a limitation of our study. However, CLAHE (and similar techniques) aims to standardize reflectance values and reduce noise in NDWI-based water detection, thus helping address challenges like varying illumination conditions and subtle spectral differences that can lead to partial misclassification of water pixels, especially at the reservoir boundary.

Another area for improvement is the development of hypsometric curves using DEM data, which is limited by the acquisition date of the DEM— with the earliest widely available dataset being the SRTM DEM (30 m) from the year 2000. Consequently, for approximately 30% of reservoirs (constructed before 2000), we utilized the recently released Global Reservoir Area-Storage-Depth Database (GRDL; Hao et al., 2024), which provides a deep learning-based bathymetry reconstruction for 7,250 GRanD reservoirs (Lehner et al., 2011), offering an alternative to traditional methods based on simplified geometric

assumptions (Hou et al., 2024; Khazaei et al., 2022; Yigzaw et al., 2018). While GRDL demonstrates superior performance compared to earlier hypsometric curve methods, its accuracy depends heavily on the size and quality of the training dataset, introducing potential uncertainties in storage estimation. Furthermore, the reproducibility of GRDL's deep learning-based results remains a challenge, limiting opportunities for further refinement and development. In contrast, geometric assumption-based methods, though less precise, offer greater flexibility and transparency for modification and advancement. While

reconstructing reservoir bathymetry remains a significant challenge, a hybrid approach that integrates geometric assumption-based methods, deep learning techniques, and field observations can yield innovative results.

Opportunity for further improvement also lies in the integration of Sentinel-1 Synthetic aperture radar (SAR) data. Unlike optical sensors, Sentinel-1 SAR can penetrate clouds and operate under all weather conditions, offering consistent and reliable

observations. The higher spatial resolution of Sentinel data (10 m) compared to Landsat (30 m) also enables more accurate classification of water and non-water pixels. Looking ahead, storage estimates can be further improved by combining Sentinel-1's microwave SAR data with observations from the recently operational Surface Water and Ocean Topography (SWOT) mission (https://swot.jpl.nasa.gov/), by the National Aeronautics and Space Administration (NASA), which provides wide coverage of water height measurements (Altenau et al., 2021; Hausman et al., 2021; Hossain et al., 2022). This integration

would not only enhance the detection and classification of water bodies but also allow for a more precise estimation of reservoir storage by linking surface area with accurate water height data.

## 6 Data and code availability

The raw satellite data used in this study were acquired from Google Earth's engine. The reservoir location information was collected from the GRanD database (Lehner et al., 2011- https://esajournals.onlinelibrary.wiley.com/doi/abs/10.1890/100125),

the Mekong database (Ang et al., 2024- https://essd.copernicus.org/articles/16/1209/2024/), and the GDAT database (Zhang and Gu, 2023- https://www.nature.com/articles/s41597-023-02008-2), which are all publicly available. The supporting data – reconstructed reservoir bathymetry – were collected from the GRDL database (Hao et al., 2024), publicly available at https://agupubs.onlinelibrary.wiley.com/doi/full/10.1029/2023WR035781. The MSEA-Res database containing the absolute reservoir storage time series and Python code is available at https://zenodo.org/records/14844580 (Mahto et al., 2025).

**Author contribution**

S.S.M., S.G., and S.F. have conceived the idea and acquired the funds. S.S.M. developed the methodology and software, conducted the investigation and analysis, curated the data, and drafted the initial manuscript. S.G., S.F., and F.H. reviewed and edited the manuscript.

**Competing interests**

The contact author has declared that none of the authors has any competing interests.

**Acknowledgments**

This work is funded by the Singapore's Ministry of Education (MoE) under its Academic Research Fund Tier 2, Project ID: MOE-000379-00/MOE-000379-01, Award Number: MOE-T2EP50122-0004. We thank NUS Singapore for providing the technical resources and infrastructure to conduct the research. We also thank Prof. Faisal Hossain for his suggestions at the

initial stages of this study.

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
