# Peer review of "A 1985-2023 time series dataset of absolute reservoir storage in Mainland Southeast Asia (MSEA-Res)"

_Earth System Science Data, 2024_

## Author Comment (AC1)

**Response to the referee comments (RCs)**

**Anonymous Referee #1**

R: The manuscript provides a long-term datasets of reservoir storage in Mainland Southeast Asia. This is meaningful for studies about the reservoir operations and further studies on hydrological processes. However, there are still some comments needed to be illustrated as listed below.

A: We thank the reviewer for the positive feedback. We will carefully address all your comments to strengthen the manuscript.

R: Line 43, Steyaert et al., 2022 and Steyaert and Condon, 2024, these two references are not found in the reference list. Please check.

A: At line 43, we referred to:

Steyaert, J. C., Condon, L. E., WD Turner, S., & Voisin, N. (2022). ResOpsUS, a dataset of historical reservoir operations in the contiguous United States. Scientific Data, 9(1), 34.

Steyaert, J. C., & Condon, L. E. (2024). Synthesis of historical reservoir operations from 1980 to 2020 for the evaluation of reservoir representation in large-scale hydrologic models. Hydrology and Earth System Sciences, 28(4), 1071-1088.

We will ensure that these two references are cited correctly in the revised version of the manuscript.

R: In table 3, for two attributes 'Water surface area (empty reservoir)' and 'Absolute storage (empty reservoir)', why they are marked as empty reservoir?

A: The water surface area and absolute storage start from a value equal to zero (i.e., representing an empty reservoir) in the Area-Elevation-Volume curves, which is why we used the expression "(empty reservoir)". We understand that this expression can be misleading, so we will remove it from the revised paper.

R: I've downloaded the dataset and am a bit confused about the water area extraction. There are three attributes 'Before_area', 'After_area' and 'Final_area', how are they derived respectively? How was the 'Final_area' determined? some of them equal to 'Before_area' and some equal to 'After_area'. Please add more explanations in the manuscript.

A: 'Before_area' refers to the water surface area of a given reservoir, calculated using the k-means classification technique applied to NDWI images. However, in binary classified images (water and non-water) with data gaps due to cloud masking, the estimated water surface area ('Before_area') is likely to be smaller than the actual water surface area. To address this, the binary images were enhanced to fill these data gaps, resulting in a revised water surface area estimate called 'After_area.' The 'Final_area' represents the area obtained after adjusting the boundary water pixels. Notably, if no adjustments were detected by the algorithm, the 'Final_area' remains equal to the 'After_area.'

To clarify these elements further, we will revise Section 3, providing a more detailed explanation of the dataset components.

R: Is the estimation improved in level 1 comparing to level 0? If so, how much?

A: Level-1 data are obtained after removing the outlier from the Level-0 data. The improvement in Level-1 compared to Level-0 varies between the reservoirs. To quantify it, we calculated the $R^2$ and nRMSE for level-0 and level-1 of the 20 reservoirs for which we have observed storage. We found that the nRMSE decreased and $R^2$ increased from Level-0 to Level-1, suggesting an improvement in our results. We will add the following supplementary figure in the revised manuscript.

[Figure]

[Figure]

R: Section 4.4, I understand that direct validation is limited by observations, but I assumed indirect validation can be applied to most of the reservoirs, why only 20 are presented? Can authors present more results?

A: For indirect validation, we need data from Altimeters, which have limited passes over the 185 reservoirs we studied here. There are 29 reservoirs for which altimeter passes are available; however, approximately 2/3$^{rd}$ of the reservoirs only have data points available for a considerable amount of time, i.e., at least ten years. Therefore, we had to limit the indirect validation to a total of 20 reservoirs. Overall, the limited availability of observations from altimeters reinforces the need to work with satellite images if one is interested in studying all / several reservoirs within a region of interest.

R: Figure 9, there are some discrepancies between the spatial distribution of precipitation deficit and storage deficit in reservoirs, especially in 2020, the lower part are not suffered from precipitation deficit but with less water stored. Please explain why.

A: The discrepancy between the spatial distribution of the precipitation and water storage anomalies is likely due to the topology of the cascading reservoir system. In other words, some reservoirs may be located in regions characterized by positive precipitation anomalies, but may

receive limited inflow from upstream reservoirs located in regions affected by droughts. We will further analyze this aspect and improve Section 4.5 to better explain Figure 9.

R: Section 4.5, drought is supposed to be a prolonged disaster that can affect the long-term water availability. It would be interesting to look into the time series of water storage in reservoirs to explore how reservoirs are affected and how they can alleviate the influence of droughts.

A: Thanks for the comment. We will include a supplementary figure showing the time series of water storage of some major reservoirs to discuss how reservoirs were affected and how they alleviated the influence of droughts in 2019-2020.

R: Line 424-426, are there any evidences that China held back water in its dams from any data or references? If not, please remove this sentence.

A: We will remove this sentence from the revised manuscript.

---

## Author Comment (AC2)

**Response to the referee comments (RCs)**

**Anonymous Referee #2**

**Edward Park**

R: I suggest a moderate revision for this manuscript. The article is well-formulated and addresses an academically significant topic with a technically robust methodology. The authors are recognized experts in this domain, and the methods employed are sound. The manuscript tackles the critical issue of reservoir constructions, compiling a valuable database of storage changes in one of the global hotspots, with significant implications for water resource management and the global population in Southeast Asia. The study is timely and has the potential to serve as an important resource for both the scientific community and policymakers working on water-related issues in the region.

Given the nature of the journal, ESSD, where the emphasis is on "resource/data publication," innovation may not necessarily be the highest priority. Instead, the value lies in building a platform that supports future research utilizing this data. Since the methods presented are sound, my comments are primarily focused on strengthening the narrative and justifying the study's broader scientific and practical implications.

A stronger justification of the research gap would enhance the manuscript. The current gaps presented seem incremental rather than innovative, improving on existing models, datasets, or studies rather than breaking new ground. While incremental research is meaningful, the study could benefit from highlighting novel techniques or scientific insights.

A: We thank the reviewer for the positive feedback. We will carefully address all your comments to further strengthen the manuscript.

R: On line 72, the paragraph starts with a question. Instead, I suggest stating the research question in a more formal manner to improve clarity.

A: We will rephrase the opening of this paragraph by clearly stating the research question.

R: On line 103, "dams" should likely be replaced with "reservoirs" for consistency and accuracy in terminology.

A: Yes—thank you for spotting this inconsistency.

R: Section 4 stands out as particularly interesting and potentially valuable. The authors provide insights into how storage patterns have evolved over the years and across different basins. This part of the study could serve as critical baseline information for future research in this domain.

R: The final section is also commendable, as it validates the utility of the database and demonstrates its application with a specific recent example. The analysis of the impact of the 2019–2020 drought on surface water storage effectively highlights the significant effects of extreme dry weather events on water resources in Mainland Southeast Asia. The demonstration of MSEA-Res's utility for hydrological modeling and other applications adds significant value to the manuscript.

A: Thank you for your positive and encouraging feedbacks.

R: Regarding figures, Fig. 1a would be more informative if the river network were included on the map. This would provide additional spatial context for readers.

A: Please find below the revised version of the figure that we plan to include in new version of the manuscript.

[Figure]

*Figure 1:* *Spatial distribution and evolution of reservoirs in Mainland Southeast Asia. (a) Map showing reservoir storage volume (km³), where the size of the circle is proportional to the reservoir capacity while the colour represents the year of commission of the dams. (b) Basin-wise distribution of dam location (red dots), stream network, and order. (c) Number of dams built per year and their corresponding cumulative storage capacity. (d) Basin-wise total number of reservoirs built until 2023.*

R: For Fig. 3, it would be helpful to include an example map or image alongside the text description for each static component. This would improve clarity and accessibility for readers unfamiliar with the methodology.

A: We also believe this addition would improve clarity and accessibility for readers. The static component (only the Area-Storage-Elevation Curve) is illustrated in Figure 4, which we have updated to show the maximum water extent and frequency maps, and thus, keeping Figure 3 unchanged. The updated Figure 4 is attached below for your reference.

[Figure]

**Figure 4:** *Illustration of the static components of MSEA-Res database (Area-Elevation-Storage relationship) for seven reservoirs, one in each of the major river basins, based on their maximum storage capacity. In each panel, (top) Elevation-Area (E-A) and Elevation-Storage (E-S) curves are shown in green and blue, respectively; (bottom-left) maximum water extent map, and (bottom-right) frequency map. The dates refer to the years of commission of the reservoirs.*

R: On line 367, the authors removed three low-performing reservoirs to improve the correlation. It would be beneficial to provide a clear justification for why these reservoirs were excluded from the statistics. Additionally, addressing the reasons behind the underperformance of certain reservoirs compared to others, as well as discussing the overall accuracy of the dataset, would strengthen the manuscript. For example, Fig. 8A suggests a potential systematic spatial distribution of $R^2$ values. If this is indeed the case, it may imply a methodological bias, which should be addressed in the discussion.

A: Our point was to emphasize that the majority of the selected reservoirs (17 out of 20) showed good agreement (average $R^2 = 0.68$ and average nRMSE = 17%) between estimated and directly observed storage, while only three (out of 20) did not agree well. This said, we understand that the current version of the paragraph could be misleading, so we will revise it to also include the average $R^2$ and nRMSE for all 20 reservoirs.

In the revised manuscript, we will also discuss the reasons behind the underperformance of certain reservoirs, which is likely due to the combination of two factors, namely:

(1) potential inaccuracies in the hypsometric curves used to transform inferred water surface into (inferred) absolute reservoir storage, and

(2) The quality of satellite data (cloud free data availability and gap filling) for NDWI estimation, which can be enhanced by satellite image pre-processing such as contrast stretching and histogram equalization.

Please also note that, to further prove the reliability of our data, we will compare the derived maps (maximum water extent and frequency) with the other data products, such as the Global Surface Water Dataset (GSWD) (Pekel et al., 2016). This is a comparison that was recommended by reviewer #3.

**References:**

1. Pekel, J. F., Cottam, A., Gorelick, N., & Belward, A. S. (2016). High-resolution mapping of global surface water and its long-term changes. Nature, 540(7633), 418-422.

---

## Author Comment (AC3)

**Response to the referee comments (RCs)**

**Anonymous Referee #3**

R: This manuscript employs Landsat and Sentinel-2 imagery to calculate the Normalized Difference Water Index (NDWI) and estimate changes in reservoir water surface area across Mainland Southeast Asia. The authors then use hypsometric curves to estimate absolute water storage dynamics for these reservoirs. These storage estimates are validated against several in-situ datasets. Using this dataset, the authors demonstrate the impact of the recent 2019-2020 drought on reservoir storage in the region. Overall, the manuscript is well-written. However, I think the use of NDWI to map surface water and the application of established hypsometric curves for storage estimation do not contribute any significant methodological innovation. I also believe it is unacceptable for a manuscript to lack a Discussion section. I also have a few major concerns outlined below:,

A: We thank the reviewer for the feedback. We will carefully address all comments to strengthen the manuscript.

We agree that mapping water surface area using NDWI is not novel, but we would also like to note that, in this study, we have not simply used NDWI images (which might be affected by cloud coverage). Rather, we corrected the NDWI images using a novel approach to address the cloud-affected areas and thus get the complete boundary of water reservoirs (please refer to Section 3). Moreover, we have integrated the enhanced NDWI images with a novel bathymetry dataset (Hao et al., 2024) to infer the reservoir's absolute storage. Finally, we would like to stress that Earth System Science Data focuses "on original research data (sets), furthering the reuse of high-quality data of benefit to Earth system sciences", rather than novel methodologies. When introducing our contribution, we thus focussed more on gaps pertaining to the existing datasets (instead of the methodologies with which they were designed); hence the reduced emphasis on our methodological approach.

As for the Discussion, please note that our discussion is embedded in Section 5 ("Conclusions"), since the journal does not provide specific guidelines on where the discussion should be placed. In the revised manuscript, we will separate 'Discussion' and 'Conclusions' and further expand the former by highlighting and discussing the comparison of our NDWI-based derived maps (i.e., maximum water extent map and frequency map – key dataset for estimating storage) with the Global Surface Water Dataset (GSWD) (Pekel et al., 2016).

R: Further validation of the surface water estimates, and hypsometric methods is necessary. I recommend comparing your surface water estimates with the Global Surface Water Dataset (GSWD) and/or other published reservoir datasets. Since you calculated water frequency rasters and maximum water extent, these can also be compared against GSWD water occurrence data to strengthen your results. Additionally, many studies have focused on developing hypsometric curves for reservoirs; it is essential to clarify why your approach is advantageous compared to others. Given the significant uncertainties in using DEMs to derive hypsometric curves, I suggest addressing these limitations in your study.

A: As suggested, we compared our water surface estimates against the ones provided by the GSWD. In particular, we began by comparing the estimates for the Sirikit and Shringarind reservoirs, which are part of the direct validation exercise (Figure 7 in the main manuscript). As shown below, the results of this comparison show a good agreement between our maps and the GSWD ones. For the revised manuscript, we will extend the comparison to all reservoirs within our database. The figures reported below (plus the additional ones we will generate) will be added to the supplementary material.

Please also note that our comparison is carried out in terms of maximum reservoir extent since one-third of the GSWD dataset is affected by gaps and is also available at a monthly frequency (Hao et al., 2024). This contrasts against our dataset, which has a sub-monthly resolution.

Finally, we will also clearly discuss the limitations of developing the hypsometric curves for reservoirs using DEMs. Specifically, we will add the following text to the revised manuscript.

*"Developing hypsometric curves using DEM data is constrained by the acquisition date of the DEM, with the earliest widely available dataset being the SRTM DEM (30 m) (year 2000). Consequently, for approximately 30% of reservoirs (constructed before 2000), we utilized the recently released Global Reservoir Area-Storage-Depth Database (GRDL; Hao et al., 2024), which employs deep learning-based bathymetry reconstruction. This database provides reliable bathymetry information for 7,250 GRanD reservoirs (Lehner et al., 2011) worldwide, offering an alternative to traditional methods based on simplified geometric assumptions (Hou et al., 2024; Khazaei et al., 2022; Yigzaw et al., 2018).*

*While GRDL demonstrates superior performance compared to earlier hypsometric curve methods, its accuracy depends heavily on the size and quality of the training dataset, introducing potential uncertainties in storage estimation. Furthermore, the reproducibility of GRDL's deep learning-based results remains a challenge, limiting opportunities for further refinement and development. In contrast, geometric assumption-based methods, though less precise, offer greater flexibility and transparency for modification and advancement."*

**Maximum reservoir extent**

**(c)** This study (283.5 km2)  **(b)** GSW (292.4 km2)

**Water occurrence (frequency) map**

**(c)** This study  **(d)** GSW

***Figure S4.*** *Comparison of maximum water extent and frequency maps with Global Surface Water Dataset (GSWD) for Sirikit reservoir.*

**Maximum reservoir extent**

**(a)** This study (413.3 km2)   **(b)** GSW (440.2 km2)

**Water occurrence (frequency) map**

**(c)** This study   **(d)** GSW

*Figure S5. Comparison of maximum water extent and frequency maps with Global Surface Water Dataset (GSWD) for Srinagarind reservoir.*

R: The use of NDWI alone may be too simplistic and may lack the accuracy needed to effectively map surface water dynamics. Without additional processing, NDWI can be prone to misclassification, especially in areas with mixed land-water pixels or seasonal vegetation cover. Additionally, factors such as high turbidity, shadows, or the presence of aquatic vegetation can further impact the accuracy of surface water mapping. Addressing these limitations is essential, and the authors might consider discussing alternative or supplementary approaches to enhance the reliability of water detection across diverse environmental conditions.

A: NDWI can be prone to misclassification, especially in areas with mixed land-water pixels or seasonal vegetation cover. This is why we have applied locally-adjusted Contrast Limited Adaptive Histogram Equalization (CLAHE, Reza, 2004) to enhance the NDWI images before classification (Section 3.3), which accounts for reducing the mixed pixel effect from the original NDWI images. Please refer to Section 3.3.

In the revised version of the manuscript, we will further discuss about techniques and opportunities to improve the reliability of the water surface estimates.

R: Combining Landsat and Sentinel-2 data should enhance the observation frequency for monitoring surface water dynamics. However, the manuscript does not highlight this potential benefit. You mention that sub-monthly surface water observations are achievable with these datasets, but it seems likely that an even higher frequency could be attained by fully leveraging both satellite sources. I recommend clarifying the observation frequency achieved in this study and discussing how the combined use of Landsat and Sentinel-2 could improve temporal resolution, potentially down to a weekly or even more frequent basis, which would provide greater detail on surface water changes.

A: Thank you for your insightful thought. We have also thought of combining a series of individual Landsat and Sentinel-2 images to generate surface water maps potentially up to a bi-weekly time period. However, many reservoirs do not fit in a single tile of Landsat and/or Sentinel-2, because of the shape and location of the reservoir. This leads to many no-data (missing) pixels over the reservoir, making image enhancement and gap-filling more challenging, and sometimes unrealistic. Therefore, we decided not to go for individual tiles of Landsat and Sentinel-2. Rather, we compromised slightly with the temporal frequency of the images and opted for using the image composites at 10-day intervals.

We will also discuss on the selection of image composite instead of individual Landsat and Sentinel-2 images in our revised manuscript.

Specific Comments:

R: Abstract: The abstract needs to be revised. It does not clearly convey that this study utilizes remote sensing data to estimate reservoir water area dynamics. Instead, it reads more like a compilation of reservoir data in Mainland Southeast Asia.

A: Thanks for your suggestion. We will modify the abstract accordingly.

R: L87: The GloLakes database provides absolute water storage data from 1984 to the present, rather than just up to 2020.

A: Thanks for spotting this inconsistency, which we will correct in the revised manuscript.

R: L101: By combining Landsat and Sentinel-2 data, it is possible to derive sub-weekly reservoir dynamics time series, offering higher temporal resolution than the sub-monthly intervals mentioned in your study.

A: We have clarified above the reason for combining Landsat and Sentinel-2 data to get sub-monthly data instead of processing individual tiles to achieve a sub-weekly time-series data.

R: L102: Why did you choose the hypsometric curves developed by Hao et al. (2024)? What advantages does this database offer over those from other studies?

A: As explained in the Introduction, the dataset developed by Hao et al. (2024) is, at this stage, the only dataset providing hypsometric curves for all 7,250 reservoirs within the GRanD. Naturally, this dataset has its own limitations, but adopting these curves is certainly a more robust approach than adopting other techniques, such as inferring the curves from a DEM and then extrapolating the curves below the water surface (Schaperow et al., 2019; Liu et al., 2020). Please also note that we did not use this dataset for all reservoirs, but only for the ones constructed before the SRTM DEM V3 was made available. That corresponds to ~30% of the reservoirs. We will strengthen this explanation in the revised manuscript.

R: Table 1: The GDAR link is not working; please check it. Additionally, the link for "Dams in the Mekong" appears to point to the GRanD database instead.

A: Thank you for spotting these errors, which we will fix.

R: Figure 1: Please ensure the volume units are consistent throughout the manuscript. "Km³" was used previously, whereas "million m³" is used here. Consider standardizing to one unit for clarity.

A: We will adopt million m$^3$ (MCM) throughout the revised manuscript.

R: L169-170: Instead of saying you "acquire" water index, water frequency, or maximum water extent, it's more accurate to state that you "derive" or "calculate" these data.

A: We will use the suggested expressions in the revised manuscript.

R: L182: The term "optical images" should not refer exclusively to "Green (G) and Near-Infrared (NIR)" bands.

A: We will rephrase the sentence in the revised manuscript as follows:

*"Shorter wavelength bands, Green (G) and Near-Infrared (NIR) can be affected by the presence of clouds – especially on rainy days – and so, NDWI."*

R: L185: Since Sentinel-2 provides a cloud cover product, have you considered using it to filter out cloudy images?

A: Yes, we have applied the cloud information to mask the raw NDWI images before making a composite in both Landsat (cloud information taken from its metadata file) and Sentinel-2.

R: L185: Images with even 5-20% cloud coverage can still significantly impact the accuracy of surface water extent measurements. This level of cloudiness may obscure key areas or introduce errors, making it essential to account for even minimal cloud presence in your analysis.

A: A 5-20% cloud coverage can still have a significant impact on the accuracy of surface water extent measurements. Therefore, to enhance the pixel accuracy, we used NDWI image composites to increase the likelihood of detecting water pixels.

R: L186: Given that Landsat has a 16-day revisit time, how are you compositing 16-day images into a 10-day interval?

A: Landsat satellites have a 16-day revisit time; however, multiple Landsat missions have often operated simultaneously (except before 1999). For example, in 2013, sensors from the Landsat-7 ETM+ and Landsat-8 series were active, enabling the creation of image composites at 10-day intervals.

R: L190-195: You need to classify these NDWI pixels before calculating water frequency and maximum water extent.

A: To clarify, we use a threshold slightly above zero (e.g., 0.1) to classify water and non-water pixels in the NDWI image. In general, a positive value (>0) indicates a water pixel, and using a higher threshold (e.g., 0.1) increases the likelihood of identifying water pixels accurately. While some water pixels with NDWI values between 0 and 0.1 might be misclassified as non-water, this effect is negligible when creating composites. By averaging more than 200 images from the Landsat and Sentinel collections (2013–2023), we estimate water frequency and maximum water extent maps with high reliability.

R: L203: "for": after?

A: Thanks for pointing this out. We will correct it.

R: L240-244: I do not understand why you need to generate level-0 data.

A: Level-0 data are the initial set of results, which are expected to have some outliers because of the uncertainty in the input NDWI images. Yes, we can only supply Level-1 and Level-2 data and not Level-0; however, the idea behind providing Level-0 data as well is to allow the users to make their own Level-1 data using other outlier removal algorithms (if required at all). We will clarify this point in the revised manuscript.

R: L246: Please specific "trend-preserving interpolation technique".

A: We intended to convey that the reservoir storage time series exhibits an increasing trend, particularly during the filling period. In such cases, linear interpolation may not perform well, especially when there are steep slopes, curvatures, or seasonal variations between data points. To address this, we employ a more robust interpolation technique, such as spline or LOESS interpolation, which fits a polynomial to the data and preserves the underlying trend.

R: Table 3: remove low dash after the words (e.g., "Level_m_")

A: We will remove them in the revised manuscript.

R: Figure 5: Do the solid lines represent Level-2 data, or are they a combination of Sentinel and Landsat data?

A: The black solid line represents the Level-2 data. We will correct the legend in our revised manuscript.

**References:**

1. Hao, Z., Chen, F., Jia, X., Cai, X., Yang, C., Du, Y., and Ling, F.: GRDL: A New Global Reservoir Area-Storage-Depth Data Set Derived Through Deep Learning-Based Bathymetry Reconstruction, Water Resources Research, 60, e2023WR035781, https://doi.org/10.1029/2023WR035781, 2024.
2. Hou, J., Van Dijk, A. I. J. M., Renzullo, L. J., and Larraondo, P. R.: GloLakes: water storage dynamics for 27000 lakes globally from 1984 to present derived from satellite altimetry and optical imaging, Earth System Science Data, 16, 201–218, https://doi.org/10.5194/essd-16-201-2024, 2024.
3. Reza, A. M.: Realization of the Contrast Limited Adaptive Histogram Equalization (CLAHE) for Real-Time Image Enhancement, The Journal of VLSI Signal Processing-Systems for Signal, Image, and Video Technology, 38, 35–44, https://doi.org/10.1023/B:VLSI.0000028532.53893.82, 2004.
4. Pekel, J. F., Cottam, A., Gorelick, N., & Belward, A. S. (2016). High-resolution mapping of global surface water and its long-term changes. Nature, 540(7633), 418-422.
5. Lehner, B., Liermann, C. R., Revenga, C., Vörösmarty, C., Fekete, B., Crouzet, P., Döll, P., Endejan, M., Frenken, K., Magome, J., Nilsson, C., Robertson, J. C., Rödel, R., Sindorf, N., and Wisser, D.: High-resolution mapping of the world's reservoirs and dams for sustainable river-flow management, Frontiers in Ecology and the Environment, 9, 494–502, https://doi.org/10.1890/100125, 2011.
6. Khazaei, B., Read, L. K., Casali, M., Sampson, K. M., and Yates, D. N.: GLOBathy, the global lakes bathymetry dataset, Sci Data, 9, 36, https://doi.org/10.1038/s41597-022-01132-9, 2022.
7. Yigzaw, W., Li, H.-Y., Demissie, Y., Hejazi, M. I., Leung, L. R., Voisin, N., and Payn, R.: A New Global Storage-Area-Depth Data Set for Modeling Reservoirs in Land Surface and Earth System Models, Water Resources Research, 54, 10,372-10,386, https://doi.org/10.1029/2017WR022040, 2018.
8. Schaperow, J. R., Li, D., Margulis, S. A., & Lettenmaier, D. P. (2019). A curve-fitting method for estimating bathymetry from water surface height and width. Water Resources Research, 55(5), 4288-4303.
9. Liu, K., Song, C., Wang, J., Ke, L., Zhu, Y., Zhu, J., ... & Luo, Z. (2020). Remote sensing-based modeling of the bathymetry and water storage for channel-type reservoirs worldwide. Water Resources Research, 56(11), e2020WR027147.

---

## Author Comment (AC4)

**Response to the referee comments (RCs)**

**Anonymous Referee #4**

This study presents an extensive database of reservoir storage timeseries for the Southeast Asian reservoirs from 1985-2023. Indeed, this paper is methodologically rigorous and can be integrated with hydrological models for the validation of their reservoir operations. However, the methodological framework section of this article needs to be significantly modified before publication to improve readability. Therefore, overall, the present version is not acceptable for publication, and I recommend a major revision for this manuscript.

A: We thank the reviewer for the positive feedback. We will carefully address all your comments to strengthen the manuscript.

Below are the comments/questions for the authors.

Abstract

1.  R: Line 15 – 16: "*The 185 reservoirs collectively store around 175 km³ (140 km³ – 210 km³) of water, covering an aggregated area of 8,700 km² (6,500 km² – 10,000 km²)*". What I understood from the manuscript is that the present total water storage (year 2023) from 185 reservoirs is 175 km$^3$. Please reflect the year in the sentence, otherwise it is confusing that which year we are referring to. Further, what is this 140 km$^3$ and 210 km$^3$ range, which I couldn't find in the manuscript? Same about the reservoir aggregated area. The area values are not described in the manuscript, especially the area range (6,500 km² – 10,000 km²). Please explain all these clearly to avoid confusion.

    A: The range inside the brackets represents the minimum and maximum values for the total storage and area in the year 2023. We will clearly mention this in our revised manuscript to avoid any further confusion.

2.  R: Line 17: "*average reservoir storage has increased from 70 km³ to 160 km³ (+130%) from 2008 to 2017*". Why the reservoir storage change from 2008 to 2017 has been considered instead of the timeframe of the database? Any specific reason for that consideration? If no, it is better to show the change from 1985 to 2023.

    A: Here, we highlighted the most significant change in reservoir storage, which occurred during the period 2008-2017, when the majority of large dams were built. Nonetheless, we will also show the change from 1985 to 2023.

3.  R: Line 18 – 21: "*Our in-situ validation provides a good match between estimated storage and in-situ observations, with 60% of the validation sites (12 out of 20) showing an R² > 0.65 and an average nRMSE < 15%. The indirect validation (based on altimetry-converted storage) shows even better results, with an R² > 0.7 and an average nRMSE < 12% for 70% (14 out of 20) of the reservoirs*". For in-situ validation, reference R$^2$ value was 0.65, whereas for indirect validation, the R$^2$ value was kept at 0.7. Please unify the reference R$^2$ value as 0.7 as like in the main text so that 10 out 20 stations (50% of the validation sites) show good agreement with in-situ observations. Rewrite the entire sentence accordingly.

A: Thank you for your suggestion. We will rewrite these sentences accordingly, keeping the $R^2$ value of 0.70 and the nRMSE value of 20% as the reference.

4. R: Line 21: "*2019-2020 drought event*". Where has this drought happened? Please specify in the sentence.

   A: The drought covered the entire region of Mainland Southeast Asia, albeit with different intensities. We will mention this point in the revised manuscript.

5. R: Line 25 – 26: "*possibility of applications in other parts of the world*". Briefly explain how this dataset will be applicable in other parts of the world in the Results section.

   A: We would like to clarify that the possibility of global applicability was in the context of the method to estimate the storage time series and not the derived dataset. We will clarify this aspect in the revised manuscript.

6. R: Line 26: "*MSEA-Res database*". This abbreviation is using for the first time in the abstract. It should be clearly explained in the earlier part of the manuscript.

   A: We will mention it clearly in the revised manuscript.

7. R: Overall: The manuscript title and abstract does not clarify regarding the methodology used to estimate the storage timeseries in MSEA. The readers need to proceed to further sections of the manuscript to understand that they have used remote sensing data to estimate the reservoir storage. It should be further emphasized in the title and abstract.

   A: Thanks for the suggestion. We will modify the title and abstract accordingly.

Introduction

1. R: Line 29 – 30: "*influencing the redistribution of water*". It should be rewritten as "influencing the distribution of water".

   A: We will modify the sentence accordingly.

2. R: Line 44: The term "Mainland Southeast Asia" has been abbreviated as MSEA in the abstract. Hence, when it appears for the first time in the Introduction, it should be again fully spelled and abbreviated, which has not been done. The MSEA abbreviation can be seen in line number 93 for the first time in Introduction without fully spelled. Also, it has been presented as "Mainland Southeast Asia" many times although it has been abbreviated. Please maintain the consistency throughout the manuscript by defining abbreviation in the first place when they appear.

   A: We will spell out the full form, 'Mainland Southeast Asia,' when introducing the abbreviation 'MSEA' for the first time in each section. While 'MSEA' is used throughout the manuscript, we have occasionally opted to use the full term 'Mainland Southeast Asia' within paragraphs to enhance readability. This approach helps prevent readers from losing track of the abbreviation, avoiding the need to revisit earlier sections for its full form, which can be distracting. In the revised manuscript, we will ensure that the use of the full form and abbreviation is consistent and systematic.

3. Line 51: "*where a few large rivers flowing in the region originate*". What the authors meant by saying "in the region originate"? Please rewrite to make it clear.

   A: Here, we meant the large rivers that originate and flow in Mainland Southeast Asia, such as the Mekong, Salween, Red, etc. We will clarify the sentence in the revised manuscript.

4. R: Line 55: Add Hanasaki et al. (2008), which is one of the pioneering works to include reservoir operation in global hydrological models. https://doi.org/10.1016/j.jhydrol.2005.11.011.

   A: Thanks. We will add the suggested reference in the revised manuscript.

5. R: Line 97 – 105: The additional data that the authors offer from Hou et al. (2024) is some reservoirs in the MSEA region with 3-year extra data from 2020. How significantly different this work is from Hou et al. (2024)? I recommend the authors to validate the storage database against the ones available in Hou et al. (2024) and discuss further whether they are comparable or not. If different, please explain why it has happened. If similar, again explanations are needed on why this new dataset is relevant.

   A: We have mentioned briefly in the introduction section on the need for our dataset over Hou et al. (2024). Please see the following paragraph:

   "*L87-95: Although Hou et al. (2024) cover the entire globe by providing a comprehensive dataset for large-scale assessments, it has a few limitations for the reservoirs located in Mainland Southeast Asia. First, the model parameters (used in the storage estimation) strongly depend on mean depth (extrapolating the surrounding topographical slope towards the centre of the lake to estimate lake depth), the surface area of the lake (derived from Landsat satellite images), and average slope (derived from DEM). Therefore, uncertainties in the estimates of reservoir storage may be generated by the estimation of depth, slope, and other model coefficients. Second, GloLakes does not include some of the largest reservoirs in MSEA, including Nuozhadu (22 km3), Xiaowan (15 km3), Xe Kaman 1 (4 km3), and Lower Seasan 2 (6 km3), which play a significant role in water redistribution and hydropower generation (Ang et al., 2024; Galelli et al., 2022; Vu et al., 2022).*"

   Nonetheless, we will validate the storage database against the ones available in Hou et al. (2024) and include them in the revised manuscript.

6. R: Line 107: 2019-2020 drought has been mentioned here as well. Clearly specify where this drought has happened.

   A: The drought covered the entire region of Mainland Southeast Asia, but with different intensities.

Water reservoirs in Mainland Southeast Asia

1. R: Table 1: the GDAT and dams in the Mekong links are not working. Please check it.

   A: Thanks for pointing this out. We will check and update this in the revised manuscript.

2. R: Figure 1: From Fig. 1(a), what I understood is that both Bhumibol and Sirikit reservoirs in the Chao Phraya basin have a reservoir storage ranging between 9000-13000 km$^3$. However, this is not true for Bhumibol, whose is storage is greater than 13000 km$^3$. Be careful about the storage capacities of all the reservoirs. Besides, Pasak dam is missing in the same basin that is included in the GRanD database. Any reason for leaving out this reservoir?

   A: We will reclassify the storage ranges in Figure 1a for better representation. You are right, the storage capacity of Bhumibol is 13,462 MCM, which is slightly higher than 13,000 MCM.

   Thank you for bringing this important detail to our attention. We sincerely appreciate your observation and will include the Pasak Dam in our revised database. It seems we may have inadvertently overlooked it during the data download process, even though it is part of the GRanD database (Lehner et al., 2011). We are grateful for your feedback and will address this in the updated version.

3. R: Figure 1 caption (Line 146): "*Basin-wise distribution of dam location (red dots), stream network, and order*". It should be rewritten as "Basin-wise distribution of dam location (red dots), stream network in the respective catchments, and stream order".

   A: Thanks for the suggestion. We will modify the caption in the revised manuscript.

Methodological Framework

1. R: Line 156: What the authors meant by sub-monthly here? How many times the data is available in a month? Please describe the minimum and maximum based on all the catchments.

   A: With the expression "sub-monthly" we referred to a 10-day interval period. Therefore, the data are available three times (i.e. three images) per month. We will clarify this point in the revised paper.

2. R: Line 156: It has been mentioned that the authors have used both Landsat and Sentinel-2 satellite data. However, the spatial resolution of Sentinel-2 is 10m and hence uncertainty and accuracy attribution will be very different for both imageries. How did the authors solve this issue? What treatment has done for the Sentinel-2 data to match its spatial resolution as that of Landsat?

   A: We used the surface reflectance bands from Landsat and Sentinel-2 to estimate NDWI, which are atmospherically-corrected (and ready-to-use) by NASA and ESA, respectively. As for the spatial resolution, we have re-gridded (with bilinear interpolation) the 10 m Sentinel-2 images to 30 m resolution, to make them consistent with the Landsat resolution. All the images (including DEM) are processed at 30 m resolution. There might be some uncertainty and accuracy mismatch between reflectance bands in Landsat and Sentinel-2.

   R: Figure 2: The reservoir maximum extent is not an input data, instead, a derived product after processing. What about the NDWI images? Is it acquired or derived? In some parts of the manuscript, it is mentioned that the NDWI has been acquired, while in some parts it says as derived. Please clearly state which way has been used to get

NDWI images. If the study uses both acquired and derived NDWI images, clearly state that when and where the derived products has used. Change Fig. 2 accordingly. Also show the steps of methodological framework in Fig. 2 for better understanding.

A: We derived the reservoir's maximum extent and NDWI images from the GEE environment. We will clearly mention in our revised manuscript that the maximum water extent, frequency map, and NDWI images are derived, whereas DEM is acquired (downloaded) using GEE. Since these three datasets are primarily required for storage estimation, we kept them in the input data ('Inputs') category.

3. R: Line 164 – 165: What is the meaning of time series satellite images? Does it mean a series of imageries?

   A: You are right. Here 'time series satellite images' means a (time) series of satellite images in a given period. We will revise the text accordingly.

4. R: Line 169 – 170: What is water frequency raster and maximum water extent raster? What is their meaning? In the later stage of manuscript, I understood that these are derived products. Then, why the authors have mentioned them as acquired input dataset?

   A: For each reservoir, the water frequency (FREQ) raster is a 30 m gridded image that indicates the probability of the presence of water (%) at each grid (or pixel) in a given period. The maximum water extent (EXT) raster is a 30 m gridded image indicating the maximum spread of water in the reservoir. Please note that the FREQ, EXT, DEM, and NDWI images are of the same dimension and geographically referenced to the same reservoir.

   We will correct the term from 'acquired' to 'derived' for FREQ, EXT, and NDWI.

5. R: Line 175 – 176: Need further information that which satellites products have used to create NDWI, water frequency raster and maximum water extent raster. Add this information in Table 2.

   A: Thanks for the suggestion. We mentioned in Lines 180-183 that Green and NIR bands are used to estimate NDWI followed by water frequency raster and maximum water extent raster. We will now add the band specifications for different satellite products in Table 2.

6. R: Line 180: The bands (green, red, and NIR) changes with satellite products and hence the NDWI formula. How to generalise these bands for all sensors? Please rewrite correctly.

   A: The NDWI formula is based on Green and NIR reflectance bands (wavelength range), which are generalized in nature. Since we are specifically mentioning the type of wavelength range (i.e. Green and NIR in this case) and not the band number, the NDWI formula still holds true and works for different satellite sensors.

7. R: Line 185: Choosing images with cloud coverage less than 80% is not a good idea. What if the cloud coverage is 89% and is exactly over the reservoir extent? What information the authors can access from such an image and how do you treat the

image further? I suggest the authors to further mask the satellite image for the reservoir extent and apply the cloud coverage threshold, preferably below 20%.

A: Thanks for the suggestion. We actually did something similar to what you suggested. The 80% cloud threshold is applied to the entire image. In the image pre-processing step, we further masked the satellite images (less than 80% cloud-affected area) for the reservoir extent and applied the cloud coverage threshold below 20% for estimating the frequency raster. We also filled the cloud-masked portion of the satellite image (NDWI) before applying the water classification algorithm. We better described these steps in the revised manuscript.

8.  R: Line 186: The revisit time of Landsat is generally 16 days. So, how did the authors make composites at 10-day intervals? Is it achieved by combing Landsat 9 and Landsat 8, whose combined temporal resolution is 8 days at the mid-latitudes. Such explanations are missing in the manuscript. I presume there could be some months without any data. How did the authors generate data for those months?

    A: Landsat has a 16-day revisit time; however, more than one Landsat mission has been active in the time domain (except for the pre-1999 period). For instance, 2013 will have active sensors from the Landsat-7 ETM+ and Landsat-8 series of satellites, making it possible to achieve image composite at an interval of 10 days.

    Yes, you are right. There could be some months without any data (Level-0 and Level-1); we generated data for those months by interpolation (Level-2). We will further elaborate on this in the revised manuscript.

    R: Line 193: How the composite NDWI has been created? For example, if we have three NDWI images with a grid cell having values of 0, 1, and 0. What will be the composite value? Is the FREQ for that particular grid 33.3, which means one-third of the grid is covered with water? Please clearly explain this in the main text or in the supplement. Further, I do not understand the EXT layer calculation. How is it calculated? How to derive the largest extent of ones from binary NDWI images? What I understood is a single FREQ and EXT maps are created for the entire period (2013-2023). Is it true? All such technical details should be clarified in the text with further details.

    A: Let us clarify this point: The NDWI composite is the average of NDWI images in a given time interval (10 days in our case). For example, if we have three NDWI images with a grid cell having values of 0, 1, and 0, then the NDWI value in the composite image will be 0.33. Please note that there can be a maximum of three composite images in each month.

    Yes, you are right. The FREQ value of the grid cell having values of 0, 1, and 0 will be 33.3. Please note that there can be only one FREQ raster (image), which is derived by averaging all the binary NDWI images (cloud percentage <20%) available over the reservoir.

    On the other hand, the EXT layer is created by taking the largest extent of ones in all binary NDWI images available between 2013 and 2023. For example, if we have three NDWI images with a grid cell having values of 0, 1, and 0, then the EXT value

will be 1 for that grid. We will clearly mention these technical details in the revised manuscript.

9. R: Line 194: To generate the NDWI, the authors used a cloud coverage threshold below 80%. But in later stages, a threshold of 20% was used to derive FREQ and EXT maps from NDWI images. Why it has to be different? Further, how the NDWI images have cloud coverage because I suppose it is already a processed image after cloud coverage removal?

A: The NDWI images with 80% or less cloud coverage are being processed for filling and correction before estimating the water surface area. In contrast, only high-quality NDWI images (with a cloud coverage threshold of 20% or less) have been used to derive the FREQ and EXT maps.

Removing cloud coverage requires masking the cloud pixels with a no-data value. In the subsequent steps, the cloud-marked (no-data) pixels are classified as either water or non-water (binary) using the FREQ and EXT rasters.

10. R: Line 197: What is scene-based NDWI image?

A: We estimated the water surface area using NDWI images, which represent individual scenes from satellite imagery. This is why we refer to them as scene-based NDWI images.

11. R: Line 190: Clearly rewrite the entire paragraph.

A: We will rewrite the paragraph for greater clarity, particularly incorporating the following explanation in the revised manuscript:

"*The NDWI composite represents the average of NDWI values over a specified time interval (10 days in our case). For example, if three NDWI images have a grid cell with values of 0, 1, and 0, the NDWI value in the composite image will be 0.33, and the FREQ value for that grid cell will be 33.3%. On the other hand, the EXT layer is created by identifying the largest extent of '1s' across all binary NDWI images. For instance, if three NDWI images have a grid cell with values of 0, 1, and 0, the EXT value for that grid cell will be 1.*"

12. R: Line 200: Did the authors compare the derived area-elevation-storage curve against observed curve? I believe it is very crucial to validate these curves because they form the heart of this study.

A: Unfortunately, no observed data are available for the area-elevation-storage curve. However, since these curves are derived from the DEM, we can still confidently rely on them, particularly for reservoirs constructed after the acquisition of the DEM (i.e., on or after the year 2000). Note that about 70% of the reservoirs fall in this category. We also note that it is common practice to derive the area-elevation-storage curve using DEM (e.g., Vu et al. 2022; Li et al., 2023). On the other hand, some uncertainty is expected with the reference area-elevation-storage curve obtained from Hao et al. (2024), which is used for reservoirs built before the year 2000 and accounted for only 30% of the reservoirs.

13. R: Line 213: Why have the authors taken the A-E-S curves from Hao et al. (2024)? Does this dataset have any specific advantages over other existing datasets?

A: We specifically used the A-E-S curves from Hao et al. (2024) for reservoirs built before the year 2000, as DEM data were not available prior to 2000. Before Hao et al. (2024), reservoir bathymetry (and thus A-E-S curves) was typically derived using geostatistical modeling approaches based on simplified geometrical assumptions and/or higher-order extrapolation techniques (Hou et al., 2024; Khazaei et al., 2022). These methods were more challenging and prone to greater uncertainty, especially when applied to a group of reservoirs with varying characteristics. Another option could be to extrapolate the hypsometric curve below the water surface (Schaperow et al., 2019; Liu et al., 2020), which may not be very reliable. The other option is to use other datasets of bathymetry, but Hao is the only one covering all dams in the GranD. Therefore, we adopted a more robust hypsometric database derived using deep learning-based bathymetry reconstruction (Hao et al., 2024).

14. R: Line 217: Is this the water surface area when the reservoir is full or area timeseries?

A: This is the time series of water surface area derived from different NDWI images. There may be some instances when the water surface corresponds to the full reservoir level.

15. R: Line 222: Why CLAHE operates in a small region? How to choose this operational window?

A: We selected CLAHE and determined the size of its operational window (8 x 8) based on the literature (Asghar et al., 2023), which suggests that CLAHE enhances the contrast and texture features of water, thereby improving the visualization of satellite images. This enhancement facilitates the classification of water and non-water pixels.

We will also add the above texts in the revised manuscript as a justification for using CLAHE.

16. R: Line 223: How is the surface area calculated? How the k-means clustering is useful here? It is not clear. Further explanations are needed.

A: We used an unsupervised classification technique (k-means clustering) to classify water pixels from NDWI images. This method does not require any training data for operation. During the pre-processing steps, each image was masked (assigned an arbitrary value, such as -1) to represent the maximum reservoir extent. In the NDWI images, higher (brighter) values indicate water. By setting the number of clusters (k) to three, the clusters correspond to water (highest cluster mean), non-water, and no data (cluster mean = -1). The area of the cluster with the highest cluster mean represents the water surface area of the reservoir. We will include additional details in the manuscript to explain how the surface area is calculated. You can also refer to Vu et al., (2022) for more details, which is the reference for developing our methodology.

Reference: Vu, D. T., Dang, T. D., Galelli, S., and Hossain, F.: Satellite observations reveal 13 years of reservoir filling strategies, operating rules, and hydrological

*alterations in the Upper Mekong River basin, Hydrology and Earth System Sciences, 26, 2345–2364, https://doi.org/10.5194/hess-26-2345-2022, 2022.*

17. R: Line 245: How the Level-2 data is generated? It has been mentioned that using a trend-preserving interpolation technique. What is it?

    A: Level-2 data are generated using the interpolation technique. Instead of using linear interpolation, we have used the 'spline' interpolation technique, which fits a non-linear function to the data, thus preserving the curvature between two points (trend-preserving interpolation technique). To avoid potential confusion, we will remove the term 'trend-preserving' and refer to it simply as the 'spline' interpolation technique.

18. R: Overall: The overall clarity and logical flow are missing in this section. Further rewriting with clear explanations on the technical details are needed.

    A: Thank you. We will improve the overall clarity of this section, and will add the suggested technical details in our revised manuscript.

Results

1. R: Line 290: In Table 3, it has been mentioned as area-level-storage, while in text wrote as area-elevation-storage. Please unify.

   A: Thank you for pointing this out. We will uniformly use the term 'area-elevation-storage' in our revised manuscript.

2. R: Line 291: The mean sea level the authors are referring to is a common datum or different for different regions.

   A: Here, the mean sea level is a common datum (i.e. WGS 1984).

3. R: Figure 4: What is the maximum storage capacity of Sirikit? In my knowledge it is blow 10000 km$^3$. Then, why the y-axis of Fig. 4(e) shows a maximum above 15000 km$^3$? The figure caption says the relationship is based on their maximum storage capacity, which is not true for at least Sirikit reservoir.

   A: The curve shown in Figure 4 also includes values beyond the maximum reservoir level (outside the reservoir). These higher values were deliberately retained to assess the uncertainty in storage estimates in case values exceeding the maximum storage capacity were encountered. You are correct that the full reservoir capacity of Sirikit is approximately 10 km³ (or 10,000 MCM), which is achieved at an elevation of ~174 m above MSL. We will update the figure caption accordingly.

4. R: Figure 5: Why the storage pattern is different for Longjing, Son La, and Nuozhadu reservoirs? They show storage fluctuations before commissioning unlike the Xe Kaman 1 reservoir.

   A: Yes, you are correct. The storage fluctuations in the Longjing, Son La, and Nuozhadu reservoirs before their commissioning are due to the limitations of the outlier removal algorithm, which should ideally align with the pattern observed in Xe Kaman 1. However, since our primary focus is on the storage time series after the

reservoirs' commissioning, our results remain reliable for drawing meaningful insights into reservoir storage dynamics and their impacts. Moreover, we provide level-1 and 2 data to allow users to clean the raw time series with the tools that seem more useful to their applications.

5. R: Figure 5: Is this a daily timeseries of sub-monthly? Which product has been used (level-1 or level-2) to plot this figure. Please mention it in the figure caption.

   A: We have used level-1 (orange and green circle) and level-2 (black-line) to plot Figure 5. We will update the Figure 5 caption accordingly in the revised manuscript.

6. R: Figure 5 caption: What is scene-based reservoir storage (Line 322)? "dynamic components" should be rewritten as "dynamic component" (Line 321).

   A: 'Scene-based reservoir storage' is the storage corresponding to the area estimated using individual scenes from satellite imagery. Since we have shown three (Level-0, Level-1, and Level-2) types of storage time series, we have used 'dynamic components' instead of 'dynamic component'.

7. R: Line 304: How to say that the Longjing (2010) reservoir was filled in roughly one year from Fig. 5(a)? What is the full capacity of that reservoir?

   A: The storage volume of the Longjing Reservoir (commissioned in 2010) was close to zero before 2010. After about a year, the reservoir began operating normally, with storage fluctuations ranging between ~0.45 km³ and ~1.25 km³, as shown in the storage time series (Figure 5a). Note that it has not dropped below 0.4 km³ since then. This indicates that the Longjing Reservoir was filled in approximately one year. The full capacity of the reservoir is 1.22 km³.

8. R: Line 327: Why to use level-1 data when the authors have a level-2 data? The authors use level-1 data in the subsequent sections as well. Any reason for this?

   A: Level-2 data are interpolated to a daily timescale, which introduces additional uncertainty when used for analyses where average storage statistics are sufficient. Therefore, we used Level-1 data instead of Level-2 data for reservoir analysis in the subsequent sections.

9. R: Line 345: It is not the volume reduction in Chao Phraya basin. Instead, the storage has substantially reduced due to persisting drought conditions and both Bhumibol and Sirikit reservoirs showed a continuous decline in storage.

   A: Thank you for your point. We will rewrite this accordingly. Specifically, we will include the following lines in the revised manuscript.

   "*In fact, the storage volume in Chao Phraya has been found to be substantially reduced by ~15% in the post-2010 due to persisting drought conditions during which both Bhumibol and Sirikit reservoirs showed a continuous decline in storage (Fig. S2b, Fig 5e)*".

10. R: Please clearly mention the temporal scale of all timeseries figures (Fig. 5, Fig. 6, Fig. 7, and Fig. 8).

A: We will clearly mention the temporal scale in the Figure captions of Fig. 5, Fig. 6, Fig. 7, and Fig. 8.

11. R: Line 371: We need to refer Fig. S2, not Fig. S3.

A: Thank you for pointing this out. We will replace Fig. S2 with Fig. S3 at Line 371.

12. R: Table S1: How the authors can explain the underperformance of Bhumibol, Rajaprbha, and Bang Lang reservoirs? Also, what is the allowable level of nRMSE to be good?

A: There are several possible explanations for the underperformance observed in the Bhumibol, Rajaprabha, and Bang Lang reservoirs. One key factor is the quality of the NDWI images used to estimate the water surface area of the reservoirs. Additionally, the presence of shallow or muddy water during dry years or when reservoir levels are low can affect the accurate classification of water pixels, leading to unrealistic estimates of reservoir storage. While this is somewhat subjective, we can consider an nRMSE of less than 20% to be acceptable, which really depends on the specific application at hand.

13. R: Line 412: Why has the reference period set between 2017 and 2023? Drought is a slow process sometimes persists for decades. Hence, the reference period has to be changed.

A: From Figure 6h, we can see that only after the 2017 the total storage reached a somewhat 'stable' value—a fact simply explained by the slowdown in the construction of new reservoirs. For this reason, we selected the period 2017–2023 to analyze the impact of drought on reservoir storage under a 'stable reservoir system' across the basins.

14. R: Line 418 – 419: storage conditions were worsened in 2020 because of the combined effect of reduced precipitation and reduced storage levels in the dams in 2019.

A: Thanks. We will rewrite the sentence accordingly in the revised manuscript.

Conclusions

1. R: Line 438: Aggregated storage capacity of nearly 175 km³ was observed by the year 2023. Please mention it.

A: We will modify the sentence accordingly in the revised manuscript.

2. R: Careful about the usage of abbreviations such as SRTM, DEM, GDAT, SAR, NASA SWOT, etc.

A: Thank you. We will mention the full form before using any abbreviations such as SRTM, DEM, GDAT, SAR, NASA SWOT, etc.

3. R: Line 459: ", flood control" should be rewritten as "and flood control".

A: We will rewrite the sentence accordingly in the revised manuscript.

**References:**

1. Hao, Z., Chen, F., Jia, X., Cai, X., Yang, C., Du, Y., and Ling, F.: GRDL: A New Global Reservoir Area-Storage-Depth Data Set Derived Through Deep Learning-Based Bathymetry Reconstruction, Water Resources Research, 60, e2023WR035781, https://doi.org/10.1029/2023WR035781, 2024.

2. Hou, J., Van Dijk, A. I. J. M., Renzullo, L. J., and Larraondo, P. R.: GloLakes: water storage dynamics for 27000 lakes globally from 1984 to present derived from satellite altimetry and optical imaging, Earth System Science Data, 16, 201–218, https://doi.org/10.5194/essd-16-201-2024, 2024.

3. Ang, W. J., Park, E., Pokhrel, Y., Tran, D. D., and Loc, H. H.: Dams in the Mekong: a comprehensive database, spatiotemporal distribution, and hydropower potentials, Earth System Science Data, 16, 1209–1228, https://doi.org/10.5194/essd-16-1209-2024, 2024.

4. Galelli, S., Dang, T. D., Ng, J. Y., Chowdhury, A. F. M. K., and Arias, M. E.: Opportunities to curb hydrological alterations via dam re-operation in the Mekong, Nat Sustain, 5, 1058–1069, https://doi.org/10.1038/s41893-022-00971-z, 2022.

5. Asghar, S., Gilanie, G., Saddique, M., Ullah, H., Mohamed, H. G., Abbasi, I. A., & Abbas, M. (2023). Water Classification Using Convolutional Neural Network. IEEE Access.

6. Lehner, B., Liermann, C. R., Revenga, C., Vörösmarty, C., Fekete, B., Crouzet, P., Döll, P., Endejan, M., Frenken, K., Magome, J., Nilsson, C., Robertson, J. C., Rödel, R., Sindorf, N., and Wisser, D.: High-resolution mapping of the world's reservoirs and dams for sustainable river-flow management, Frontiers in Ecology and the Environment, 9, 494–502, https://doi.org/10.1890/100125, 2011.

7. Khazaei, B., Read, L. K., Casali, M., Sampson, K. M., and Yates, D. N.: GLOBathy, the global lakes bathymetry dataset, Sci Data, 9, 36, https://doi.org/10.1038/s41597-022-01132-9, 2022.

8. Schaperow, J. R., Li, D., Margulis, S. A., & Lettenmaier, D. P. (2019). A curve-fitting method for estimating bathymetry from water surface height and width. Water Resources Research, 55(5), 4288-4303.

9. Liu, K., Song, C., Wang, J., Ke, L., Zhu, Y., Zhu, J., ... & Luo, Z. (2020). Remote sensing-based modeling of the bathymetry and water storage for channel-type reservoirs worldwide. Water Resources Research, 56(11), e2020WR027147.

10. Li, Y., Zhao, G., Allen, G. H., and Gao, H.: Diminishing storage returns of reservoir construction, Nat Commun, 14, 3203, https://doi.org/10.1038/s41467-023-38843-5, 2023.

11. Vu, D. T., Dang, T. D., Galelli, S., and Hossain, F.: Satellite observations reveal 13 years of reservoir filling strategies, operating rules, and hydrological alterations in the Upper Mekong River basin, Hydrology and Earth System Sciences, 26, 2345–2364, https://doi.org/10.5194/hess-26-2345-2022, 2022.

---

## Author Response (AR1)

**Response to the referee comments (RCs)**

**Anonymous Referee #1**

R: The manuscript provides a long-term datasets of reservoir storage in Mainland Southeast Asia. This is meaningful for studies about the reservoir operations and further studies on hydrological processes. However, there are still some comments needed to be illustrated as listed below.

A: We thank the reviewer for the positive feedback. We have carefully addressed all your comments to strengthen the manuscript.

R: Line 43, Steyaert et al., 2022 and Steyaert and Condon, 2024, these two references are not found in the reference list. Please check.

A: At line 43, we referred to:

Steyaert, J. C., Condon, L. E., WD Turner, S., & Voisin, N. (2022). ResOpsUS, a dataset of historical reservoir operations in the contiguous United States. Scientific Data, 9(1), 34.

Steyaert, J. C., & Condon, L. E. (2024). Synthesis of historical reservoir operations from 1980 to 2020 for the evaluation of reservoir representation in large-scale hydrologic models. Hydrology and Earth System Sciences, 28(4), 1071-1088.

We have now cited the above-mentioned references in the revised manuscript.

R: In table 3, for two attributes 'Water surface area (empty reservoir)' and 'Absolute storage (empty reservoir)', why they are marked as empty reservoir?

A: The water surface area and absolute storage start from a value equal to zero (i.e., representing an empty reservoir) in the Area-Elevation-Volume curves, which is why we used the expression "(empty reservoir)". We understand that this expression can be misleading, so we have removed it.

R: I've downloaded the dataset and am a bit confused about the water area extraction. There are three attributes 'Before_area', 'After_area' and 'Final_area', how are they derived respectively? How was the 'Final_area' determined? some of them equal to 'Before_area' and some equal to 'After_area'. Please add more explanations in the manuscript.

A: 'Before_area' refers to the water surface area of a given reservoir, calculated using the k-means classification technique applied to NDWI images. However, in binary classified images (water and non-water) with data gaps due to cloud masking, the estimated water surface area ('Before_area') is likely to be smaller than the actual water surface area. To address this, the binary images were enhanced to fill these data gaps, resulting in a revised water surface area estimate called 'After_area.' The 'Final_area' represents the area obtained after adjusting the boundary water pixels. Notably, if no adjustments were detected by the algorithm, the 'Final_area' remains equal to the 'After_area.'

To clarify these elements further, we have revised Section 3.3, providing a more detailed explanation of the dataset components.

Particularly, we have added the following sentences in the revised manuscript:

Line 259-262:

*'To find NDWI thresholds for each satellite image, we resort to k-means clustering. Eventually, the preliminary water pixels were identified by selecting the cluster corresponding to the maximum centroid value of NDWI. The water surface area estimated from the preliminary water pixels is referred to as 'Before_area' for any given reservoir (Table 3).'*

Line 2267-271:

*'We add the clear water pixels (k-means clustering) and cloud-filled water pixels (Vu et al., 2022) to get a complete picture of the reservoir water surface area for each NDWI image, called 'After_area' (Table 3). Finally, we adjust the boundary water pixels of the complete reservoir water surface area, represented as 'Final_area'. Note that, if no adjustments were detected by the algorithm, the 'Final_area' remains equal to the 'After_area' (Table 3).'*

R: Is the estimation improved in level 1 comparing to level 0? If so, how much?

A: Level-1 data are obtained after removing the outlier from the Level-0 data. The improvement in Level-1 compared to Level-0 varies between the reservoirs. To quantify it, we calculated the $R^2$ and nRMSE for level-0 and level-1 of the 20 reservoirs for which we have observed storage. We found that the nRMSE decreased and $R^2$ increased from Level-0 to Level-1, suggesting that the outlier removal process can further enhance the quality of the data.

We have added the following figure in the revised manuscript

1. Supplementary figure

[Figure]

[Figure]

*Figure S1: nRMSE and R² between inferred (Level-0 and Level-1) and observed data. The nRMSE decreased and R² increased from Level-0 to Level-1, suggesting an improvement in the outlier removal approach.*

2. Main text

Line 275-285:

*'For each reservoir, Level-0 corresponds to the scene-based (instantaneous) raw outputs of absolute reservoir storage, which have been derived from the available satellite images. We then performed a simple box plot analysis on Level-0 data to remove the outliers, creating the so-called Level-1 data. Note that these data are created using a generalized box-plot framework for quality control that is not specifically designed for each reservoir; therefore, on a case-to-case basis, some values in the storage time series may still be considered outliers—they can be removed manually or with the aid of other data analysis algorithms. Therefore, the improvement in Level-1 data compared to Level-0 data varies between the reservoirs. To quantify it, we calculated the $R^2$ and nRMSE for level-0 and level-1 data of the 20 reservoirs for which we have the observed storage. The detailed analysis on the 20 selected reservoirs is presented below in section 4.4 (Figure 7 and Table S2). We found that the nRMSE decreased and $R^2$ increased from Level-0 to Level-1, suggesting that the outlier removal process can further enhance the quality of the data (Figure S4).'*

R: Section 4.4, I understand that direct validation is limited by observations, but I assumed indirect validation can be applied to most of the reservoirs, why only 20 are presented? Can authors present more results?

A: For indirect validation, we need data from Altimeters, which have limited passes over the 185 reservoirs we studied here. There are 29 reservoirs for which altimeter passes are available; however, approximately 2/3[rd] of the reservoirs only have data points available for a considerable amount of time, i.e., at least ten years. Therefore, we had to limit the indirect validation to a total of 20 reservoirs. Overall, the limited availability of observations from altimeters reinforces the need to work with satellite images if one is interested in studying all / several reservoirs within a region of interest.

R: Figure 9, there are some discrepancies between the spatial distribution of precipitation deficit and storage deficit in reservoirs, especially in 2020, the lower part are not suffered from precipitation deficit but with less water stored. Please explain why.

A: The discrepancy between the spatial distribution of the precipitation and water storage anomalies is likely due to the topology of the cascading reservoir system. In other words, some reservoirs may be located in regions characterized by positive precipitation anomalies, but may receive limited inflow from upstream reservoirs located in regions affected by droughts.

We have added the above justification to Section 4.5 to better explain Figure 9. Particularly we have added,

Line 470-475:

*'Storage conditions were worsened in 2020, with 144 of 186 reservoirs (78%) exhibiting negative storage departures, primarily due to the combined effects of precipitation deficits in both 2019 and 2020 (Fig. 9d). Interestingly, we noticed some discrepancy between the spatial distribution of the precipitation and water storage anomalies (Fig. 9), likely due to the topology of the cascading reservoir system. In other words, some reservoirs located in regions characterized by positive precipitation anomalies, but may receive limited inflow from upstream reservoirs located in regions affected by droughts.'*

R: Section 4.5, drought is supposed to be a prolonged disaster that can affect the long-term water availability. It would be interesting to look into the time series of water storage in reservoirs to explore how reservoirs are affected and how they can alleviate the influence of droughts.

A: Thanks for the comment. The time series of water storage of some reservoirs (20 reservoirs each for direct and indirect validation) in Figures S3 and S4 clearly shows the effect of 2019-2020 drought in reservoir's storage capacity in that period.

R: Line 424-426, are there any evidences that China held back water in its dams from any data or references? If not, please remove this sentence.

A: We have removed the sentence from the revised manuscript.

**Response to the referee comments (RCs)**

**Anonymous Referee #2**

**Edward Park**

R: I suggest a moderate revision for this manuscript. The article is well-formulated and addresses an academically significant topic with a technically robust methodology. The authors are recognized experts in this domain, and the methods employed are sound. The manuscript tackles the critical issue of reservoir constructions, compiling a valuable database of storage changes in one of the global hotspots, with significant implications for water resource management and the global population in Southeast Asia. The study is timely and has the potential to serve as an important resource for both the scientific community and policymakers working on water-related issues in the region.

Given the nature of the journal, ESSD, where the emphasis is on "resource/data publication," innovation may not necessarily be the highest priority. Instead, the value lies in building a platform that supports future research utilizing this data. Since the methods presented are sound, my comments are primarily focused on strengthening the narrative and justifying the study's broader scientific and practical implications.

A stronger justification of the research gap would enhance the manuscript. The current gaps presented seem incremental rather than innovative, improving on existing models, datasets, or studies rather than breaking new ground. While incremental research is meaningful, the study could benefit from highlighting novel techniques or scientific insights.

A: We thank the reviewer for the positive feedback. We have now carefully addressed all your comments to strengthen the manuscript.

R: On line 72, the paragraph starts with a question. Instead, I suggest stating the research question in a more formal manner to improve clarity.

A: Thank you for your valuable suggestion. I have revised the paragraph to improve clarity by rephrasing the research question in a more formal manner. The updated text avoids starting with a direct question while maintaining the intended meaning.

R: On line 103, "dams" should likely be replaced with "reservoirs" for consistency and accuracy in terminology.

A: Yes—thank you for spotting this inconsistency. We have replaced 'dams' with 'reservoirs' wherever possible, except in cases where the original term is necessary.

R: Section 4 stands out as particularly interesting and potentially valuable. The authors provide insights into how storage patterns have evolved over the years and across different basins. This part of the study could serve as critical baseline information for future research in this domain.

R: The final section is also commendable, as it validates the utility of the database and demonstrates its application with a specific recent example. The analysis of the impact of the 2019–2020 drought on surface water storage effectively highlights the significant effects of

extreme dry weather events on water resources in Mainland Southeast Asia. The demonstration of MSEA-Res's utility for hydrological modeling and other applications adds significant value to the manuscript.

A: Thank you for your positive and encouraging feedback.

R: Regarding figures, Fig. 1a would be more informative if the river network were included on the map. This would provide additional spatial context for readers.

A: We have updated Figure 1a accordingly (attached below) in the revised manuscript.

[Figure]

*Figure 1:* Spatial distribution and evolution of reservoirs in Mainland Southeast Asia. (a) Map showing reservoir storage volume (km³), where the size of the circle is proportional to the reservoir capacity while the colour represents the year of commission of the dams. (b) Basin-wise distribution of dam location (red dots), stream network, and order. (c) Number of dams built per year and their corresponding cumulative storage capacity. (d) Basin-wise total number of reservoirs built until 2023.

R: For Fig. 3, it would be helpful to include an example map or image alongside the text description for each static component. This would improve clarity and accessibility for readers unfamiliar with the methodology.

A: We also believe this addition would improve clarity and accessibility for readers. The static component (only the Area-Elevation-Storage Curve) is illustrated in Figure 4, which we have

updated to show the maximum water extent and frequency maps, thus, keeping Figure 3 unchanged. Accordingly, we have updated Figure 4 (attached below) in the revised manuscript.

[Figure]

***Figure 4:*** *Illustration of the static components of MSEA-Res database (Area-Elevation-Storage relationship) for seven reservoirs, one in each of the major river basins. In each panel, (top) Elevation-Area (E-A) and Elevation-Storage (E-S) curves are shown in green and blue, respectively; (bottom-left) maximum water extent map, and (bottom-right) frequency map. The dates refer to the years of commission of the reservoirs.*

R: On line 367, the authors removed three low-performing reservoirs to improve the correlation. It would be beneficial to provide a clear justification for why these reservoirs were excluded from the statistics. Additionally, addressing the reasons behind the underperformance of certain reservoirs compared to others, as well as discussing the overall accuracy of the dataset, would strengthen the manuscript. For example, Fig. 8A suggests a potential systematic spatial distribution of $R^2$ values. If this is indeed the case, it may imply a methodological bias, which should be addressed in the discussion.

A: Our point was to emphasize that the majority of the selected reservoirs (17 out of 20) showed good agreement (average $R^2 = 0.68$ and average nRMSE = 17%) between estimated and directly observed storage, while only three (out of 20) did not agree well. This said, we understand that the current version of the paragraph could be misleading, so we have revised it to also include the average $R^2$ and nRMSE for all 20 reservoirs.

In the revised manuscript, we have added the following text:

Line 413-414:

*"As expected, the average $R^2$ and nRMSE across all 20 reservoirs are approximately 0.6 and 18.6%, respectively (Table S1)."*

Line 426-433:

*"The underperformance of a few reservoirs can likely be attributed to two key factors. First, potential inaccuracies in the hypsometric curves may introduce errors when converting inferred water surface area into absolute reservoir storage. Second, the quality of satellite-derived NDWI data, particularly cloud-free image availability and gap filling, can significantly impact accuracy. Enhancing satellite image pre-processing through techniques such as contrast stretching and histogram equalization could improve data quality and, in turn, refine reservoir storage estimations. Addressing these challenges will be crucial in further optimizing the framework's reliability across diverse hydrological settings. Despite these challenges, the direct and indirect validation metrics suggest that the InfeRes-derived storage data can be reliably used for water storage-related analysis on a weekly to yearly time scale."*

**Please also note that to further prove the reliability of our data, we have also compared the derived maps (maximum water extent and frequency) with the other data products, such as the Global Surface Water Dataset (GSWD) (Pekel et al., 2016). This is a comparison that was recommended by reviewer #3, which is also attached below.**

We compared our water surface estimates against the ones provided by the GSWD. In particular, we began by comparing the estimates for Sirikit and Shringarind reservoirs, which are part of the direct validation exercise (Figure 7 in the main manuscript). As shown below, the results of this comparison show a good agreement between our maps and the GSWD ones. For the revised manuscript, we have extended the comparison (with GSWD) of maximum water extent of all reservoirs within our database, which overall showed an excellent agreement ($R^2 = 0.98$) across the 186 reservoirs. The figures reported below (plus the additional ones we generated) have been added to the supplementary material.

Please also note that our comparison is carried out in terms of maximum reservoir extent since one-third of the GSWD dataset is affected by gaps and is available at a monthly frequency (Hao et al., 2024). This contrasts against our dataset, which has a sub-monthly resolution.

[Figure]

**Figure S3.** *Comparison of maximum water extent and frequency maps with Global Surface Water Dataset (GSWD) for Sirikit reservoir.*

[Figure]

**Figure S2.** *Comparison of maximum water extent and frequency maps with Global Surface Water Dataset (GSWD) for Srinagarind reservoir.*

[Figure]

***Figure S1.*** *Comparison of maximum water extent with Global Surface Water Dataset (GSWD) for 186 reservoirs across the Mainland Southeast Asia.*

**References:**

1.  Pekel, J. F., Cottam, A., Gorelick, N., & Belward, A. S. (2016). High-resolution mapping of global surface water and its long-term changes. Nature, 540(7633), 418-422.

**Response to the referee comments (RCs)**

**Anonymous Referee #3**

R: This manuscript employs Landsat and Sentinel-2 imagery to calculate the Normalized Difference Water Index (NDWI) and estimate changes in reservoir water surface area across Mainland Southeast Asia. The authors then use hypsometric curves to estimate absolute water storage dynamics for these reservoirs. These storage estimates are validated against several in-situ datasets. Using this dataset, the authors demonstrate the impact of the recent 2019-2020 drought on reservoir storage in the region. Overall, the manuscript is well-written. However, I think the use of NDWI to map surface water and the application of established hypsometric curves for storage estimation do not contribute any significant methodological innovation. I also believe it is unacceptable for a manuscript to lack a Discussion section. I also have a few major concerns outlined below:,

A: We thank the reviewer for the feedback. We have now carefully address all comments to strengthen the manuscript.

We agree that mapping water surface area using NDWI is not fully novel, but we would also like to note that, in this study, we have not simply used NDWI images (which might be affected by cloud coverage). Rather, we corrected the NDWI images using a novel approach to address the cloud-affected areas and thus get the complete boundary of water reservoirs (please refer to Section 3). Moreover, we have integrated the enhanced NDWI images with a novel bathymetry dataset (Hao et al., 2024) to infer the reservoir's absolute storage. Finally, we would like to stress that Earth System Science Data focuses "on original research data (sets), furthering the reuse of high-quality data of benefit to Earth system sciences", rather than novel methodologies. When introducing our contribution, we thus focussed more on gaps pertaining to the existing datasets (instead of the methodologies with which they were designed); hence the reduced emphasis on our methodological approach.

As for the Discussion, please note that our discussion is embedded in Section 5 ("Conclusions"), since the journal does not provide specific guidelines on where the discussion should be placed. In the revised manuscript, we have renamed Section 5 ("Discussion and Conclusions") to discuss the scope of various image processing techniques, development of hypsometric curves, and integrating multi-satellite datasets to further enhance the water surface estimates. We also highlighted the comparison of our NDWI-based derived maps (i.e., maximum water extent map, a key dataset for estimating storage) with the Global Surface Water Dataset (GSWD) (Pekel et al., 2016).

R: Further validation of the surface water estimates, and hypsometric methods is necessary. I recommend comparing your surface water estimates with the Global Surface Water Dataset (GSWD) and/or other published reservoir datasets. Since you calculated water frequency rasters and maximum water extent, these can also be compared against GSWD water occurrence data to strengthen your results. Additionally, many studies have focused on developing hypsometric curves for reservoirs; it is essential to clarify why your approach is advantageous compared to

others. Given the significant uncertainties in using DEMs to derive hypsometric curves, I suggest addressing these limitations in your study.

A: As suggested, we compared our water surface estimates against the ones provided by the GSWD. In particular, we began by comparing the estimates for Sirikit and Shringarind reservoirs, which are part of the direct validation exercise (Figure 7 in the main manuscript). As shown below, the results of this comparison show a good agreement between our maps and the GSWD ones. For the revised manuscript, we have extended the comparison (with GSWD) of maximum water extent of all reservoirs within our database, which overall showed an excellent agreement ($R^2 = 0.98$) across the 186 reservoirs. The figures reported below (plus the additional ones we generated) have been added to the supplementary material.

Please also note that our comparison is carried out in terms of maximum reservoir extent since one-third of the GSWD dataset is affected by gaps and is available at a monthly frequency (Hao et al., 2024). This contrasts against our dataset, which has a sub-monthly resolution.

[Figure]

*Figure S3. Comparison of maximum water extent and frequency maps with Global Surface Water Dataset (GSWD) for Sirikit reservoir.*

[Figure]

*Figure S2. Comparison of maximum water extent and frequency maps with Global Surface Water Dataset (GSWD) for Srinagarind reservoir.*

[Figure]

*Figure S1. Comparison of maximum water extent with Global Surface Water Dataset (GSWD) for 186 reservoirs across the Mainland Southeast Asia.*

Finally, we also discussed the limitations of developing the hypsometric curves for reservoirs using DEMs. Specifically, we added the following text to the revised manuscript.

Line 531-542:

*"Another area for improvement is the development of hypsometric curves using DEM data, which is limited by the acquisition date of the DEM— with the earliest widely available dataset being the SRTM DEM (30 m) from the year 2000. Consequently, for approximately 30% of reservoirs (constructed before 2000), we utilized the recently released Global Reservoir Area-Storage-Depth Database (GRDL; Hao et al., 2024), which provides deep learning-based bathymetry reconstruction for 7,250 GRanD reservoirs (Lehner et al., 2011), offering an alternative to traditional methods based on simplified geometric assumptions (Hou et al., 2024; Khazaei et al., 2022; Yigzaw et al., 2018). While GRDL demonstrates superior performance compared to earlier hypsometric curve methods, its accuracy depends heavily on the size and quality of the training dataset, introducing potential uncertainties in storage estimation. Furthermore, the reproducibility of GRDL's deep learning-based results remains a challenge, limiting opportunities for further refinement and development. In contrast, geometric assumption-based methods, though less precise, offer greater flexibility and transparency for modification and advancement. While reconstructing reservoir bathymetry remains a significant challenge, a hybrid approach that integrates geometric assumption-based methods, deep learning techniques, and field observations can yield innovative results."*

R: The use of NDWI alone may be too simplistic and may lack the accuracy needed to effectively map surface water dynamics. Without additional processing, NDWI can be prone to misclassification, especially in areas with mixed land-water pixels or seasonal vegetation cover. Additionally, factors such as high turbidity, shadows, or the presence of aquatic vegetation can further impact the accuracy of surface water mapping. Addressing these limitations is essential, and the authors might consider discussing alternative or supplementary approaches to enhance the reliability of water detection across diverse environmental conditions.

A: NDWI can be prone to misclassification, especially in areas with mixed land-water pixels or seasonal vegetation cover. This is why we have applied locally-adjusted Contrast Limited Adaptive Histogram Equalization (CLAHE, Reza, 2004) to enhance the NDWI images before classification (Section 3.3), which accounts for reducing the mixed pixel effect from the original NDWI images. Please refer to Section 3.3.

R: Combining Landsat and Sentinel-2 data should enhance the observation frequency for monitoring surface water dynamics. However, the manuscript does not highlight this potential benefit. You mention that sub-monthly surface water observations are achievable with these datasets, but it seems likely that an even higher frequency could be attained by fully leveraging both satellite sources. I recommend clarifying the observation frequency achieved in this study and discussing how the combined use of Landsat and Sentinel-2 could improve temporal resolution, potentially down to a weekly or even more frequent basis, which would provide greater detail on surface water changes.

A: Thank you for your insightful thought. We have also thought of combining a series of individual Landsat and Sentinel-2 images to generate surface water maps potentially up to a bi-weekly time period. However, many reservoirs do not fit in a single tile of Landsat and/or Sentinel-2, because of the shape and location of the reservoir. This leads to many no-data (missing) pixels over the reservoir, making image enhancement and gap-filling more challenging, and sometimes unrealistic. Therefore, we decided not to go for individual tiles of Landsat and Sentinel-2. Rather, we compromised slightly with the temporal frequency of the images and opted for using the image composites at 10-day intervals.

Specific Comments:

R: Abstract: The abstract needs to be revised. It does not clearly convey that this study utilizes remote sensing data to estimate reservoir water area dynamics. Instead, it reads more like a compilation of reservoir data in Mainland Southeast Asia.

A: Thanks for your suggestion. We have added the following sentence in the revised manuscript:

Line 13-15:

*"This dataset is derived from remote sensing observations, integrating satellite-based water surface area extraction from high-resolution (30m) images and Area-Elevation-Storage (AES) relationships to estimate reservoir level and storage dynamics."*

R: L87: The GloLakes database provides absolute water storage data from 1984 to the present, rather than just up to 2020.

A: Thanks for spotting this inconsistency, which we have now corrected.

R: L101: By combining Landsat and Sentinel-2 data, it is possible to derive sub-weekly reservoir dynamics time series, offering higher temporal resolution than the sub-monthly intervals mentioned in your study.

A: We have clarified above the reason for combining Landsat and Sentinel-2 data to get sub-monthly data instead of processing individual tiles to achieve a sub-weekly time-series data.

R: L102: Why did you choose the hypsometric curves developed by Hao et al. (2024)? What advantages does this database offer over those from other studies?

A: As explained in the Introduction, the dataset developed by Hao et al. (2024) is, at this stage, the only dataset providing hypsometric curves for all 7,250 reservoirs within the GRanD. Naturally, this dataset has its own limitations, but adopting these curves is certainly a more robust approach than adopting other techniques, such as inferring the curves from a DEM and then extrapolating the curves below the water surface (Schaperow et al., 2019; Liu et al., 2020). Please also note that we did not use this dataset for all reservoirs, but only for the ones constructed before the SRTM DEM V3 was made available. That corresponds to ~30% of the reservoirs.

Particularly, we have added the following texts:

Line 531-542:

*"The development of hypsometric curves using DEM data, is limited by the acquisition date of the DEM— with the earliest widely available dataset being the SRTM DEM (30 m) from the year 2000. Consequently, for approximately 30% of reservoirs (constructed before 2000), we utilized the recently released Global Reservoir Area-Storage-Depth Database (GRDL; Hao et al., 2024), which provides deep learning-based bathymetry reconstruction for 7,250 GRanD reservoirs (Lehner et al., 2011), offering an alternative to traditional methods based on simplified geometric assumptions (Hou et al., 2024; Khazaei et al., 2022; Yigzaw et al., 2018). While GRDL demonstrates superior performance compared to earlier hypsometric curve methods, its accuracy depends heavily on the size and quality of the training dataset, introducing potential uncertainties in storage estimation. Furthermore, the reproducibility of GRDL's deep learning-based results remains a challenge, limiting opportunities for further refinement and development. In contrast, geometric assumption-based methods, though less precise, offer greater flexibility and transparency for modification and advancement. While reconstructing reservoir bathymetry remains a significant challenge, a hybrid approach that integrates geometric assumption-based methods, deep learning techniques, and field observations can yield innovative results."*

R: Table 1: The GDAR link is not working; please check it. Additionally, the link for "Dams in the Mekong" appears to point to the GRanD database instead.

A: Thank you for spotting these errors, which we have fixed.

R: Figure 1: Please ensure the volume units are consistent throughout the manuscript. "Km³" was used previously, whereas "million m³" is used here. Consider standardizing to one unit for clarity.

A: We have adopted $km^3$ (cubic kilometres) throughout the revised manuscript.

R: L169-170: Instead of saying you "acquire" water index, water frequency, or maximum water extent, it's more accurate to state that you "derive" or "calculate" these data.

A: We have replaced 'acquire' with 'derived' in the revised manuscript.

R: L182: The term "optical images" should not refer exclusively to "Green (G) and Near-Infrared (NIR)" bands.

A: We have rephrased the sentence in the revised manuscript as follows:

Line 195-196:

*"Shorter wavelength bands, Green (G) and Near-Infrared (NIR) can be affected by the presence of clouds – especially on rainy days – and so, NDWI."*

R: L185: Since Sentinel-2 provides a cloud cover product, have you considered using it to filter out cloudy images?

A: Yes, we have applied the cloud information to mask the raw NDWI images before making a composite in both Landsat (cloud information taken from its metadata file) and Sentinel-2.

R: L185: Images with even 5-20% cloud coverage can still significantly impact the accuracy of surface water extent measurements. This level of cloudiness may obscure key areas or introduce errors, making it essential to account for even minimal cloud presence in your analysis.

A: A 5-20% cloud coverage can still have a significant impact on the accuracy of surface water extent measurements. Therefore, to enhance the pixel accuracy, we used NDWI image composites to increase the likelihood of detecting water pixels.

R: L186: Given that Landsat has a 16-day revisit time, how are you compositing 16-day images into a 10-day interval?

A: Landsat satellites have a 16-day revisit time; however, multiple Landsat missions have often operated simultaneously (except before 1999). For example, in 2013, sensors from the Landsat-7 ETM+ and Landsat-8 series were active, enabling the creation of image composites at 10-day intervals.

R: L190-195: You need to classify these NDWI pixels before calculating water frequency and maximum water extent.

A: To clarify, we use a threshold slightly above zero (e.g., 0.1) to classify water and non-water pixels in the NDWI image. In general, a positive value (>0) indicates a water pixel, and using a higher threshold (e.g., 0.1) increases the likelihood of identifying water pixels accurately. While some water pixels with NDWI values between 0 and 0.1 might be misclassified as non-water, this effect is negligible when creating composites. By averaging more than 200 images from the Landsat and Sentinel collections (2013–2023), we estimate water frequency and maximum water extent maps with high reliability.

R: L203: "for": after?

A: Thanks for pointing this out. We have corrected it.

R: L240-244: I do not understand why you need to generate level-0 data.

A: Level-0 data are the initial set of results, which are expected to have some outliers because of the uncertainty in the input NDWI images. Yes, we can only supply Level-1 and Level-2 data and not Level-0; however, the idea behind providing Level-0 data as well is to allow the users to make their own Level-1 data using other outlier removal algorithms (if required at all). We have clarified this point in the revised manuscript.

Particularly, we added:

Line 275-278:

*"For each reservoir, Level-0 corresponds to the scene-based (instantaneous) raw outputs of absolute reservoir storage, which have been derived from the available satellite images. We then performed a simple box plot analysis on Level-0 data to remove the outliers, creating the so-*

*called Level-1 data. Level-0 data are provided to give users the flexibility to generate their own Level-1 data using alternative outlier removal algorithms, if needed."*

R: L246: Please specific "trend-preserving interpolation technique".

A: We intended to convey that the reservoir storage time series exhibits an increasing trend, particularly during the filling period. In such cases, linear interpolation may not perform well, especially when there are steep slopes, curvatures, or seasonal variations between data points. To address this, we employ a more robust interpolation technique, such as spline or LOESS interpolation, which fits a polynomial to the data and preserves the underlying trend. We have replaced "*trend-preserving interpolation technique*" to '*non-linear (i.e. spline) interpolation*' for simplification.

R: Table 3: remove low dash after the words (e.g., "Level_m_")

A: We have removed them in the revised manuscript.

R: Figure 5: Do the solid lines represent Level-2 data, or are they a combination of Sentinel and Landsat data?

A: The black solid line represents the Level-2 data. We have corrected Figure 5 accordingly.

**References:**

1. Hao, Z., Chen, F., Jia, X., Cai, X., Yang, C., Du, Y., and Ling, F.: GRDL: A New Global Reservoir Area-Storage-Depth Data Set Derived Through Deep Learning-Based Bathymetry Reconstruction, Water Resources Research, 60, e2023WR035781, https://doi.org/10.1029/2023WR035781, 2024.
2. Hou, J., Van Dijk, A. I. J. M., Renzullo, L. J., and Larraondo, P. R.: GloLakes: water storage dynamics for 27000 lakes globally from 1984 to present derived from satellite altimetry and optical imaging, Earth System Science Data, 16, 201–218, https://doi.org/10.5194/essd-16-201-2024, 2024.
3. Reza, A. M.: Realization of the Contrast Limited Adaptive Histogram Equalization (CLAHE) for Real-Time Image Enhancement, The Journal of VLSI Signal Processing-Systems for Signal, Image, and Video Technology, 38, 35–44, https://doi.org/10.1023/B:VLSI.0000028532.53893.82, 2004.
4. Pekel, J. F., Cottam, A., Gorelick, N., & Belward, A. S. (2016). High-resolution mapping of global surface water and its long-term changes. Nature, 540(7633), 418-422.
5. Lehner, B., Liermann, C. R., Revenga, C., Vörösmarty, C., Fekete, B., Crouzet, P., Döll, P., Endejan, M., Frenken, K., Magome, J., Nilsson, C., Robertson, J. C., Rödel, R., Sindorf, N., and Wisser, D.: High-resolution mapping of the world's reservoirs and dams for sustainable river-flow management, Frontiers in Ecology and the Environment, 9, 494–502, https://doi.org/10.1890/100125, 2011.

6.  Khazaei, B., Read, L. K., Casali, M., Sampson, K. M., and Yates, D. N.: GLOBathy, the global lakes bathymetry dataset, Sci Data, 9, 36, https://doi.org/10.1038/s41597-022-01132-9, 2022.
7.  Yigzaw, W., Li, H.-Y., Demissie, Y., Hejazi, M. I., Leung, L. R., Voisin, N., and Payn, R.: A New Global Storage-Area-Depth Data Set for Modeling Reservoirs in Land Surface and Earth System Models, Water Resources Research, 54, 10,372-10,386, https://doi.org/10.1029/2017WR022040, 2018.
8.  Schaperow, J. R., Li, D., Margulis, S. A., & Lettenmaier, D. P. (2019). A curve-fitting method for estimating bathymetry from water surface height and width. Water Resources Research, 55(5), 4288-4303.
9.  Liu, K., Song, C., Wang, J., Ke, L., Zhu, Y., Zhu, J., ... & Luo, Z. (2020). Remote sensing-based modeling of the bathymetry and water storage for channel-type reservoirs worldwide. Water Resources Research, 56(11), e2020WR027147.

**Response to the referee comments (RCs)**

**Anonymous Referee #4**

This study presents an extensive database of reservoir storage timeseries for the Southeast Asian reservoirs from 1985-2023. Indeed, this paper is methodologically rigorous and can be integrated with hydrological models for the validation of their reservoir operations. However, the methodological framework section of this article needs to be significantly modified before publication to improve readability. Therefore, overall, the present version is not acceptable for publication, and I recommend a major revision for this manuscript.

A: We thank the reviewer for the positive feedback. We have carefully addressed all your comments to strengthen the manuscript.

Below are the comments/questions for the authors.

Abstract

1.  R: Line 15 – 16: "*The 185 reservoirs collectively store around 175 km³ (140 km³ – 210 km³) of water, covering an aggregated area of 8,700 km² (6,500 km² – 10,000 km²)*". What I understood from the manuscript is that the present total water storage (year 2023) from 185 reservoirs is 175 km$^3$. Please reflect the year in the sentence, otherwise it is confusing that which year we are referring to. Further, what is this 140 km$^3$ and 210 km$^3$ range, which I couldn't find in the manuscript? Same about the reservoir aggregated area. The area values are not described in the manuscript, especially the area range (6,500 km² – 10,000 km²). Please explain all these clearly to avoid confusion.

    A: The range inside the brackets represents the minimum and maximum values for the total storage and area in the year 2023. We have clearly mentioned this in our revised manuscript to avoid any further confusion. Particularly we have added:

    Line 17-19:

    *"The 186 reservoirs collectively store around 175 km³ of water, with a minimum of 140 km³ and a maximum of 210 km³. They cover an aggregated area of 8,700 km², ranging from a minimum of 6,500 km² to a maximum of 10,000 km²."*

2.  R: Line 17: "*average reservoir storage has increased from 70 km³ to 160 km³ (+130%) from 2008 to 2017*". Why the reservoir storage change from 2008 to 2017 has been considered instead of the timeframe of the database? Any specific reason for that consideration? If no, it is better to show the change from 1985 to 2023.

    A: Here, we highlighted the most significant change in reservoir storage, which occurred during the period 2008-2017, when the majority (~55%) of large dams were built.

3.  R: Line 18 – 21: "*Our in-situ validation provides a good match between estimated storage and in-situ observations, with 60% of the validation sites (12 out of 20) showing an R² > 0.65 and an average nRMSE < 15%. The indirect validation (based on altimetry-converted storage) shows even better results, with an R² > 0.7 and an average nRMSE < 12% for 70% (14 out of 20) of the reservoirs*". For in-situ

validation, reference $R^2$ value was 0.65, whereas for indirect validation, the $R^2$ value was kept at 0.7. Please unify the reference $R^2$ value as 0.7 as like in the main text so that 10 out 20 stations (50% of the validation sites) show good agreement with in-situ observations. Rewrite the entire sentence accordingly.

A: Thank you for your suggestion. We have revised the sentence as follows:

Line 21-24:

*"Our in-situ validation provides a good match between estimated storage and in-situ observations, with 50% of the validation sites (10 out of 20) showing an R² > 0.7 and an average nRMSE < 14%. The indirect validation (based on altimetry-converted storage) shows even better results, with an R² > 0.7 and an average nRMSE < 12% for 70% (14 out of 20) of the reservoirs."*

4. R: Line 21: "*2019-2020 drought event*". Where has this drought happened? Please specify in the sentence.

   A: We have modified the sentence as follows:

   Line 24-25:

   *"Furthermore, the analysis of the 2019-2020 drought event in the MSEA region reveals that nearly 30-40% of the region experienced more than five months of drought, with the most significant impact on reservoirs in Cambodia and Thailand."*

5. R: Line 25 – 26: "*possibility of applications in other parts of the world*". Briefly explain how this dataset will be applicable in other parts of the world in the Results section.

   A: We would like to clarify that the possibility of global applicability was in the context of the method to estimate the storage time series and not the derived dataset.

   We have modified the sentence as follows:

   Line 27-29:

   *"Overall, this analysis demonstrates the potential of the inferred storage time series for assessing real-life water-related problems in Mainland Southeast Asia, with the possibility of applying the method to estimate reservoir storage time series in other parts of the world."*

6. R: Line 26: "*MSEA-Res database*". This abbreviation is using for the first time in the abstract. It should be clearly explained in the earlier part of the manuscript.

   A: We have mentioned it clearly in the revised manuscript.

7. R: Overall: The manuscript title and abstract does not clarify regarding the methodology used to estimate the storage timeseries in MSEA. The readers need to proceed to further sections of the manuscript to understand that they have used remote sensing data to estimate the reservoir storage. It should be further emphasized in the title and abstract.

   A: Thanks for the suggestion. We have modified the abstract accordingly.

Introduction

1. R: Line 29 – 30: "*influencing the redistribution of water*". It should be rewritten as "influencing the distribution of water".

   A: Corrected.

2. R: Line 44: The term "Mainland Southeast Asia" has been abbreviated as MSEA in the abstract. Hence, when it appears for the first time in the Introduction, it should be again fully spelled and abbreviated, which has not been done. The MSEA abbreviation can be seen in line number 93 for the first time in Introduction without fully spelled. Also, it has been presented as "Mainland Southeast Asia" many times although it has been abbreviated. Please maintain the consistency throughout the manuscript by defining abbreviation in the first place when they appear.

   A: We have spelled out the full term, 'Mainland Southeast Asia,' when introducing the acronym 'MSEA' for the first time in each section. While 'MSEA' is used throughout the manuscript, we have occasionally opted to use the full term 'Mainland Southeast Asia' within paragraphs to enhance readability. This approach helps prevent readers from losing track of the abbreviation, avoiding the need to revisit earlier sections for its full form, which can be distracting.

3. Line 51: "*where a few large rivers flowing in the region originate*". What the authors meant by saying "in the region originate"? Please rewrite to make it clear.

   A: Here, we meant the large rivers that originate and flow in Mainland Southeast Asia, such as the Mekong, Salween, Red, etc.

   We have modified the sentence as follows:

   Line 52-54:

   *"With this concern in mind, we focus on the reservoirs of Mainland Southeast Asia, including Myanmar, Thailand, Laos, Vietnam, Cambodia, Malaysia, Singapore, and part of southern China—where several major rivers originate and flow through the region."*

4. R: Line 55: Add Hanasaki et al. (2008), which is one of the pioneering works to include reservoir operation in global hydrological models. https://doi.org/10.1016/j.jhydrol.2005.11.011.

   A: Thanks. We have added the following reference:

   *Hanasaki, N., Kanae, S., and Oki, T.: A reservoir operation scheme for global river routing models, Journal of Hydrology, 327, 22–41, https://doi.org/10.1016/j.jhydrol.2005.11.011, 2006.*

5. R: Line 97 – 105: The additional data that the authors offer from Hou et al. (2024) is some reservoirs in the MSEA region with 3-year extra data from 2020. How significantly different this work is from Hou et al. (2024)? I recommend the authors to validate the storage database against the ones available in Hou et al. (2024) and discuss further whether they are comparable or not. If different, please explain why it

has happened. If similar, again explanations are needed on why this new dataset is relevant.

A: We have mentioned briefly in the introduction section on the need for our dataset over Hou et al. (2024).

Please see the following paragraph:

Line 91-99:

*"Although Hou et al. (2024) cover the entire globe by providing a comprehensive dataset for large-scale assessments, it has a few limitations for the reservoirs located in Mainland Southeast Asia. First, the model parameters (used in the storage estimation) strongly depend on mean depth (extrapolating the surrounding topographical slope towards the centre of the lake to estimate lake depth), the surface area of the lake (derived from Landsat satellite images), and average slope (derived from DEM). Therefore, uncertainties in the estimates of reservoir storage may be generated by the estimation of depth, slope, and other model coefficients. Second, GloLakes does not include some of the largest reservoirs in MSEA, including Nuozhadu (22 km³), Xiaowan (15 km³), Xe Kaman 1 (4 km³), and Lower Seasan 2 (6 km³), which play a significant role in water redistribution and hydropower generation (Ang et al., 2024; Galelli et al., 2022; Vu et al., 2022)."*

6. R: Line 107: 2019-2020 drought has been mentioned here as well. Clearly specify where this drought has happened.

A: The drought covered the entire region of Mainland Southeast Asia, but with different intensities. We have corrected it in the revised manuscript.

Water reservoirs in Mainland Southeast Asia

1. R: Table 1: the GDAT and dams in the Mekong links are not working. Please check it.

A: Thanks for pointing this out. We have checked and updated the links in the revised manuscript.

2. R: Figure 1: From Fig. 1(a), what I understood is that both Bhumibol and Sirikit reservoirs in the Chao Phraya basin have a reservoir storage ranging between 9000-13000 km³. However, this is not true for Bhumibol, whose is storage is greater than 13000 km³. Be careful about the storage capacities of all the reservoirs. Besides, Pasak dam is missing in the same basin that is included in the GRanD database. Any reason for leaving out this reservoir?

A: Thanks. We have now reclassified the storage ranges in Figure 1a for better representation. You are right, the storage capacity of Bhumibol is 13,462 MCM, which is slightly higher than 13,000 MCM; it is clearly marked now in the updated Figure 1a. Please see the following Figure.

[Figure]

*Figure 1: Spatial distribution and evolution of reservoirs in Mainland Southeast Asia. (a) Map showing reservoir storage volume (km3), where the size of the circle is proportional to the reservoir capacity while the colour represents the year of commission of the reservoirs. (b) Basin-wise distribution of reservoir location (red dots), stream network in the respective catchments, and stream order. (c) Number of reservoirs built per year and their corresponding cumulative storage capacity. (d) Basin-wise total number of reservoirs built until 2023.*

Thank you for bringing this important detail to our attention. We sincerely appreciate your observation and have included the Pasak Dam in our revised database. It seems we may have inadvertently overlooked it during the data download process, even though it is part of the GRanD database (Lehner et al., 2011). We are grateful for your feedback and have addressed this in the updated version.

3.  R: Figure 1 caption (Line 146): "*Basin-wise distribution of dam location (red dots), stream network, and order*". It should be rewritten as "Basin-wise distribution of dam location (red dots), stream network in the respective catchments, and stream order".

    A: Thanks for the suggestion. We have modified the caption accordingly.

Methodological Framework

1. R: Line 156: What the authors meant by sub-monthly here? How many times the data is available in a month? Please describe the minimum and maximum based on all the catchments.

   A: With the expression "sub-monthly" we referred to a 10-day interval period. Therefore, the data are available three times (i.e. three images) per month. We have clarified it in the revised manuscript.

2. R: Line 156: It has been mentioned that the authors have used both Landsat and Sentinel-2 satellite data. However, the spatial resolution of Sentinel-2 is 10m and hence uncertainty and accuracy attribution will be very different for both imageries. How did the authors solve this issue? What treatment has done for the Sentinel-2 data to match its spatial resolution as that of Landsat?

   A: We used the surface reflectance bands from Landsat and Sentinel-2 to estimate NDWI, which are atmospherically-corrected (and ready-to-use) by NASA and ESA, respectively. As for the spatial resolution, we have re-gridded (with bilinear interpolation) the 10 m Sentinel-2 images to 30 m resolution, to make them consistent with the Landsat resolution. All the images (including DEM) are processed at 30 m resolution. There might be some uncertainty and accuracy mismatch between reflectance bands in Landsat and Sentinel-2 series of satellites, however, this is not crucial.

   R: Figure 2: The reservoir maximum extent is not an input data, instead, a derived product after processing. What about the NDWI images? Is it acquired or derived? In some parts of the manuscript, it is mentioned that the NDWI has been acquired, while in some parts it says as derived. Please clearly state which way has been used to get NDWI images. If the study uses both acquired and derived NDWI images, clearly state that when and where the derived products has used. Change Fig. 2 accordingly. Also show the steps of methodological framework in Fig. 2 for better understanding.

   A: We derived the reservoir's maximum extent and NDWI images from the GEE environment. We have clearly mentioned in our revised manuscript that the maximum water extent, frequency map, and NDWI images are derived, whereas DEM is acquired (downloaded) using GEE. Since these three datasets are primarily required for storage estimation, we kept them in the input data ('Inputs') category.

   We have added the following sentence in the figure (2) caption and within text:

   Line 178-179 and 183-184:

   *"Please note that the maximum water extent, frequency map, and NDWI images are the derived data, whereas DEM is acquired (downloaded) using Google Earth Engine (GEE) Python API."*

3. R: Line 164 – 165: What is the meaning of time series satellite images? Does it mean a series of imageries?

   A: You are right. Here 'time series satellite images' means a (time) series of satellite images in a given period. We have revised the text accordingly.

4. R: Line 169 – 170: What is water frequency raster and maximum water extent raster? What is their meaning? In the later stage of manuscript, I understood that these are derived products. Then, why the authors have mentioned them as acquired input dataset?

A: For each reservoir, the water frequency (FREQ) raster is a 30 m gridded image that indicates the probability of the presence of water (%) at each grid (or pixel) in a given period. The maximum water extent (EXT) raster is a 30 m gridded image indicating the maximum spread of water in the reservoir. Please note that the FREQ, EXT, DEM, and NDWI images are of the same dimension and geographically referenced to the same reservoir.

We have now corrected the term from 'acquired' to 'derived' for FREQ, EXT, and NDWI.

5. R: Line 175 – 176: Need further information that which satellites products have used to create NDWI, water frequency raster and maximum water extent raster. Add this information in Table 2.

A: Thanks for the suggestion. We mentioned that Green and NIR bands are used to estimate NDWI followed by water frequency raster and maximum water extent raster. Please see the following:

Line 193-194:

*"The Green (G) and Near-infrared (NIR) bands from the satellite sensors (Landsat and Sentinel) are used to calculate NDWI [i.e. (G-NIR)/(G+NIR)] – as proposed by McFeeters, (1996) – for the available scenes, collectively covering the study period 1985-2023."*

6. R: Line 180: The bands (green, red, and NIR) changes with satellite products and hence the NDWI formula. How to generalise these bands for all sensors? Please rewrite correctly.

A: The NDWI formula is based on Green and NIR reflectance bands (wavelength range), which are generalized in nature. Since we are specifically mentioning the type of wavelength range (i.e. Green and NIR in this case) and not the band number, the NDWI formula still holds true and works for different satellite sensors.

7. R: Line 185: Choosing images with cloud coverage less than 80% is not a good idea. What if the cloud coverage is 89% and is exactly over the reservoir extent? What information the authors can access from such an image and how do you treat the image further? I suggest the authors to further mask the satellite image for the reservoir extent and apply the cloud coverage threshold, preferably below 20%.

A: Thanks for the suggestion. We actually did something similar to what you suggested. The 80% cloud threshold is applied to the entire image. In the image pre-processing step, we further masked the satellite images (less than 80% cloud-affected area) for the reservoir extent and applied the cloud coverage threshold below 20% for estimating the frequency raster. We also filled the cloud-masked portion of the satellite image (NDWI) before applying the water classification algorithm.

8. R: Line 186: The revisit time of Landsat is generally 16 days. So, how did the authors make composites at 10-day intervals? Is it achieved by combing Landsat 9 and Landsat 8, whose combined temporal resolution is 8 days at the mid-latitudes. Such explanations are missing in the manuscript. I presume there could be some months without any data. How did the authors generate data for those months?

A: Landsat has a 16-day revisit time; however, more than one Landsat mission has been active in the time domain (except for the pre-1999 period). For instance, 2013 has active sensors from the Landsat-7 ETM+ and Landsat-8 series of satellites, making it possible to achieve image composite at an interval of 10 days.

Yes, you are right. There could be some months without any data (Level-0 and Level-1); we generated data for those months by interpolation (Level-2), which we have clearly mentioned in the revised manuscript.

Particularly we have added the following sentences:

Line 162-166:

*"Despite Landsat having a 16-day revisit time, we could achieve a 10-day interval data because more than one Landsat mission has been active in the time domain (except for the pre-1999 period). For instance, 2013 has active sensors from the Landsat-7 ETM+ and Landsat-8 series of satellites, making it possible to achieve image composite at an interval of 10 days. Please note that there could be some months without any satellite data, resulting in storage unavailability in those months, which we filled by interpolation."*

R: Line 193: How the composite NDWI has been created? For example, if we have three NDWI images with a grid cell having values of 0, 1, and 0. What will be the composite value? Is the FREQ for that particular grid 33.3, which means one-third of the grid is covered with water? Please clearly explain this in the main text or in the supplement. Further, I do not understand the EXT layer calculation. How is it calculated? How to derive the largest extent of ones from binary NDWI images? What I understood is a single FREQ and EXT maps are created for the entire period (2013-2023). Is it true? All such technical details should be clarified in the text with further details.

A: Let us clarify this point: The NDWI composite is the average of NDWI images in a given time interval (10 days in our case). For example, if we have three NDWI images with a grid cell having values of 0, 1, and 0, then the NDWI value in the composite image will be 0.33. Please note that there can be a maximum of three composite images in each month.

Yes, you are right. The FREQ value of the grid cell having values of 0, 1, and 0 will be 33.3. Please note that there can be only one FREQ raster (image), which is derived by averaging all the binary NDWI images (cloud percentage <20%) available over the reservoir.

On the other hand, the EXT layer is created by taking the largest extent of ones in all binary NDWI images available between 2013 and 2023. For example, if we have three NDWI images with a grid cell having values of 0, 1, and 0, then the EXT value

will be 1 for that grid. We have now clearly mentioned these technical details in the revised manuscript.

Please see the following:

Line 199-205:

*"We also made NDWI composites from available Landsat (1985-2023) and Sentinel (2016-2023) images at 10-day intervals, which is the average of NDWI images in a given time interval (10 days in our case). For example, if we have three NDWI images with a grid cell having values of 0, 1, and 0, then the NDWI value in the composite image will be 0.33. Please note that there can be a maximum of three composite images in each month (i.e., only from Landsat) during the period 1985-2015. On the other hand there can be a maximum of six images per month (three from Landsat and three from Sentinel) in 2016-2023. Making a composite of NDWI images maximizes the chances of getting more cloud-free pixels than individual NDWI images."*

Line 213-221:

*"The FREQ layer is created by making a composite of all binary NDWI images (more than 200 images from the Landsat and Sentinel collections), whose cloud percentage is less than 20% (i.e., clear sky condition) and by dividing it by the total number of selected images (cloud percentage <20%). We multiply the FREQ layer by 100 to get the percentage of water present at each pixel. For example, if three NDWI images make a composite image of value 0.33 at any grid, the FREQ value for that grid cell will be 33.3%. Please note that there can be only one FREQ raster (image), which is derived by averaging all the binary NDWI images (cloud percentage <20%) available over the reservoir. Subsequently, the EXT layer is created by simply taking the largest extent of ones in all binary NDWI images available between 2013 and 2023. For example, if we have three NDWI images with a grid cell having values of 0, 1, and 0, then the EXT value will be 1 for that grid."*

9. R: Line 194: To generate the NDWI, the authors used a cloud coverage threshold below 80%. But in later stages, a threshold of 20% was used to derive FREQ and EXT maps from NDWI images. Why it has to be different? Further, how the NDWI images have cloud coverage because I suppose it is already a processed image after cloud coverage removal?

A: The NDWI images with 80% or less cloud coverage are being processed for filling and correction before estimating the water surface area. In contrast, only high-quality NDWI images (with a cloud coverage threshold of 20% or less) have been used to derive the FREQ and EXT maps.

Removing cloud coverage requires masking the cloud pixels with a no-data value. In the subsequent steps, the cloud-marked (no-data) pixels are classified as either water or non-water (binary) using the FREQ and EXT rasters.

10. R: Line 197: What is scene-based NDWI image?

A: We estimated the water surface area using NDWI images, which represent individual scenes from satellite imagery. This is why we refer to them as scene-based NDWI images.

11. R: Line 190: Clearly rewrite the entire paragraph.

A: We have rewritten the paragraph for greater clarity, particularly incorporating the following explanation in the revised manuscript:

Line 199-201:

"*We also made NDWI composites from available Landsat (1985-2023) and Sentinel (2016-2023) images at 10-day intervals, which is the average of NDWI images in a given time interval (10 days in our case). For example, if we have three NDWI images with a grid cell having values of 0, 1, and 0, then the NDWI value in the composite image will be 0.33.*"

12. R: Line 200: Did the authors compare the derived area-elevation-storage curve against observed curve? I believe it is very crucial to validate these curves because they form the heart of this study.

A: Unfortunately, no observed data are available for the area-elevation-storage curve. However, since these curves are derived from the DEM, we can still confidently rely on them, particularly for reservoirs constructed after the acquisition of the DEM (i.e., on or after the year 2000). Note that about 70% of the reservoirs fall in this category. We also note that it is common practice to derive the area-elevation-storage curve using DEM (e.g., Vu et al. 2022; Li et al., 2023). On the other hand, some uncertainty is expected with the reference area-elevation-storage curve obtained from Hao et al. (2024), which is used for reservoirs built before the year 2000 and accounted for only 30% of the reservoirs.

13. R: Line 213: Why have the authors taken the A-E-S curves from Hao et al. (2024)? Does this dataset have any specific advantages over other existing datasets?

A: We specifically used the A-E-S curves from Hao et al. (2024) for reservoirs built before the year 2000, as DEM data were not available prior to 2000. Before Hao et al. (2024), reservoir bathymetry (and thus A-E-S curves) was typically derived using geostatistical modeling approaches based on simplified geometrical assumptions and/or higher-order extrapolation techniques (Hou et al., 2024; Khazaei et al., 2022). These methods were more challenging and prone to greater uncertainty, especially when applied to a group of reservoirs with varying characteristics. Another option could be to extrapolate the hypsometric curve below the water surface (Schaperow et al., 2019; Liu et al., 2020), which may not be very reliable. The other option is to use other datasets of bathymetry, but Hao is the only one covering all dams in the GranD. Therefore, we adopted a more robust hypsometric database derived using deep learning-based bathymetry reconstruction (Hao et al., 2024).

14. R: Line 217: Is this the water surface area when the reservoir is full or area timeseries?

A: This is the time series of water surface area derived from different NDWI images. There may be some instances when the water surface corresponds to the full reservoir level.

15. R: Line 222: Why CLAHE operates in a small region? How to choose this operational window?

A: We selected CLAHE and determined the size of its operational window (8 x 8) based on the literature (Asghar et al., 2023), which suggests that CLAHE enhances the contrast and texture features of water, thereby improving the visualization of satellite images. This enhancement facilitates the classification of water and non-water pixels.

We have added the following sentences in the revised manuscript to improve the technicality:

Line 250-256:

*"A locally-adjusted Contrast Limited Adaptive Histogram Equalization (CLAHE) was applied to enhance the NDWI images before classification. CLAHE (Reza, 2004) is a variant of Adaptive histogram equalization (AHE), which takes care of over-amplification of the contrast in an image. CLAHE operates on small regions in the image (8 x 8 pixels window in our case) rather than the entire image. The size of its operational window (8 x 8) is based on the literature (Asghar et al., 2023), which suggests that CLAHE enhances the contrast and texture features of water, thereby improving the visualization of satellite images. This enhancement facilitates the classification of water and non-water pixels."*

16. R: Line 223: How is the surface area calculated? How the k-means clustering is useful here? It is not clear. Further explanations are needed.

A: We used an unsupervised classification technique (k-means clustering) to classify water pixels from NDWI images. This method does not require any training data for operation. During the pre-processing steps, each image was masked (assigned an arbitrary value, such as -1) to represent the maximum reservoir extent. In the NDWI images, higher (brighter) values indicate water. By setting the number of clusters (k) to three, the clusters correspond to water (highest cluster mean), non-water, and no data (cluster mean = -1). The area of the cluster with the highest cluster mean represents the water surface area of the reservoir. We have included additional details in the manuscript to explain how the surface area is calculated. You can also refer to Vu et al., (2022) for more details, which is the reference for developing our methodology.

*Reference: Vu, D. T., Dang, T. D., Galelli, S., and Hossain, F.: Satellite observations reveal 13 years of reservoir filling strategies, operating rules, and hydrological alterations in the Upper Mekong River basin, Hydrology and Earth System Sciences, 26, 2345–2364, https://doi.org/10.5194/hess-26-2345-2022, 2022.*

17. R: Line 245: How the Level-2 data is generated? It has been mentioned that using a trend-preserving interpolation technique. What is it?

A: Level-2 data are generated using the interpolation technique. Instead of using linear interpolation, we have used the 'spline' interpolation technique, which fits a non-linear function to the data, thus preserving the curvature between two points (trend-preserving interpolation technique). To avoid potential confusion, we have removed the term 'trend-preserving' and refer to it simply as the non-linear ('spline') interpolation technique.

18. R: Overall: The overall clarity and logical flow are missing in this section. Further rewriting with clear explanations on the technical details are needed.

A: Thank you. We have incorporated all your suggestions to improve the overall clarity and technical details of the manuscript.

Results

1. R: Line 290: In Table 3, it has been mentioned as area-level-storage, while in text wrote as area-elevation-storage. Please unify.

A: Thank you for pointing this out. We have now uniformly used the term 'area-elevation-storage' in our revised manuscript.

2. R: Line 291: The mean sea level the authors are referring to is a common datum or different for different regions.

A: Here, the mean sea level is a common datum (i.e. WGS 1984).

3. R: Figure 4: What is the maximum storage capacity of Sirikit? In my knowledge it is blow 10000 km$^3$. Then, why the y-axis of Fig. 4(e) shows a maximum above 15000 km$^3$? The figure caption says the relationship is based on their maximum storage capacity, which is not true for at least Sirikit reservoir.

A: The curve shown in Figure 4 also includes values beyond the maximum reservoir level (outside the reservoir). These higher values were deliberately retained to assess the uncertainty in storage estimates in case values exceeding the maximum storage capacity were encountered. You are correct that the full reservoir capacity of Sirikit is approximately 10 km³ (or 10,000 MCM), which is achieved at an elevation of ~174 m above MSL. We have updated the figure caption accordingly.

4. R: Figure 5: Why the storage pattern is different for Longjing, Son La, and Nuozhadu reservoirs? They show storage fluctuations before commissioning unlike the Xe Kaman 1 reservoir.

A: Yes, you are correct. The storage fluctuations in the Longjing, Son La, and Nuozhadu reservoirs before their commissioning are due to the limitations of the outlier removal algorithm, which should ideally align with the pattern observed in Xe Kaman 1. However, since our primary focus is on the storage time series after the reservoirs' commissioning, our results remain reliable for drawing meaningful insights into reservoir storage dynamics and their impacts. Moreover, we provide level-1 and 2 data to allow users to clean the raw time series with the tools that seem more useful to their applications.

5. R: Figure 5: Is this a daily timeseries of sub-monthly? Which product has been used (level-1 or level-2) to plot this figure. Please mention it in the figure caption.

   A: We have used level-1 (orange and green circle) and level-2 (black-line) to plot Figure 5. We have updated the Figure 5 caption accordingly in the revised manuscript.

6. R: Figure 5 caption: What is scene-based reservoir storage (Line 322)? "dynamic components" should be rewritten as "dynamic component" (Line 321).

   A: 'Scene-based reservoir storage' is the storage corresponding to the area estimated using individual scenes from satellite imagery. Since we have shown three (Level-0, Level-1, and Level-2) types of storage time series, we have used 'dynamic components' instead of 'dynamic component'.

7. R: Line 304: How to say that the Longjing (2010) reservoir was filled in roughly one year from Fig. 5(a)? What is the full capacity of that reservoir?

   A: The storage volume of the Longjing Reservoir (commissioned in 2010) was close to zero before 2010. After about a year, the reservoir began operating normally, with storage fluctuations ranging between ~0.45 km³ and ~1.25 km³, as shown in the storage time series (Figure 5a). Note that it has not dropped below 0.4 km³ since then. This indicates that the Longjing Reservoir was filled in approximately one year. The full capacity of the reservoir is 1.22 km³.

8. R: Line 327: Why to use level-1 data when the authors have a level-2 data? The authors use level-1 data in the subsequent sections as well. Any reason for this?

   A: Level-2 data are interpolated to a daily timescale, which introduces additional uncertainty when used for analyses where average storage statistics are sufficient. Therefore, we used Level-1 data instead of Level-2 data for reservoir analysis in the subsequent sections.

9. R: Line 345: It is not the volume reduction in Chao Phraya basin. Instead, the storage has substantially reduced due to persisting drought conditions and both Bhumibol and Sirikit reservoirs showed a continuous decline in storage.

   A: Thank you for your point. We have included the following lines in the revised manuscript.

   Line 386-388:

   *"In fact, the storage volume in Chao Phraya has been found to be substantially reduced by ~15% in the post-2010 (Fig 6e), due to persisting drought conditions during which both Bhumibol and Sirikit reservoirs showed a continuous decline in storage (Fig. S6b, Fig 5e)."*

10. R: Please clearly mention the temporal scale of all timeseries figures (Fig. 5, Fig. 6, Fig. 7, and Fig. 8).

    A: We have mentioned the temporal scale in the Figure captions of Fig. 6, Fig. 7, and Fig. 8.

11. R: Line 371: We need to refer Fig. S2, not Fig. S3.

A: Thank you for pointing this out. We have corrected it now.

12. R: Table S1: How the authors can explain the underperformance of Bhumibol, Rajaprbha, and Bang Lang reservoirs? Also, what is the allowable level of nRMSE to be good?

A: There are several possible explanations for the underperformance observed in the Bhumibol, Rajaprabha, and Bang Lang reservoirs. One key factor is the quality of the NDWI images used to estimate the water surface area of the reservoirs. Additionally, the presence of shallow or muddy water during dry years or when reservoir levels are low can affect the accurate classification of water pixels, leading to unrealistic estimates of reservoir storage. While this is somewhat subjective, we can consider an nRMSE of less than 20% to be acceptable, which really depends on the specific application at hand.

13. R: Line 412: Why has the reference period set between 2017 and 2023? Drought is a slow process sometimes persists for decades. Hence, the reference period has to be changed.

A: From Figure 6h, we can see that only after 2017 the total storage reached a somewhat 'stable' value—a fact simply explained by the slowdown in the construction of new reservoirs. For this reason, we selected the period 2017–2023 to analyze the impact of drought on reservoir storage under a 'stable reservoir system' across the basins.

14. R: Line 418 – 419: storage conditions were worsened in 2020 because of the combined effect of reduced precipitation and reduced storage levels in the dams in 2019.

A: Thanks. We have rewritten the sentence accordingly in the revised manuscript.

Conclusions

1. R: Line 438: Aggregated storage capacity of nearly 175 km³ was observed by the year 2023. Please mention it.

A: We have modified the sentence accordingly.

2. R: Careful about the usage of abbreviations such as SRTM, DEM, GDAT, SAR, NASA SWOT, etc.

A: Thank you. We have mentioned it now.

3. R: Line 459: ", flood control" should be rewritten as "and flood control".

A: Done.

**References:**

1. Hao, Z., Chen, F., Jia, X., Cai, X., Yang, C., Du, Y., and Ling, F.: GRDL: A New Global Reservoir Area-Storage-Depth Data Set Derived Through Deep Learning-Based

Bathymetry Reconstruction, Water Resources Research, 60, e2023WR035781, https://doi.org/10.1029/2023WR035781, 2024.

2. Hou, J., Van Dijk, A. I. J. M., Renzullo, L. J., and Larraondo, P. R.: GloLakes: water storage dynamics for 27000 lakes globally from 1984 to present derived from satellite altimetry and optical imaging, Earth System Science Data, 16, 201–218, https://doi.org/10.5194/essd-16-201-2024, 2024.

3. Ang, W. J., Park, E., Pokhrel, Y., Tran, D. D., and Loc, H. H.: Dams in the Mekong: a comprehensive database, spatiotemporal distribution, and hydropower potentials, Earth System Science Data, 16, 1209–1228, https://doi.org/10.5194/essd-16-1209-2024, 2024.

4. Galelli, S., Dang, T. D., Ng, J. Y., Chowdhury, A. F. M. K., and Arias, M. E.: Opportunities to curb hydrological alterations via dam re-operation in the Mekong, Nat Sustain, 5, 1058–1069, https://doi.org/10.1038/s41893-022-00971-z, 2022.

5. Asghar, S., Gilanie, G., Saddique, M., Ullah, H., Mohamed, H. G., Abbasi, I. A., & Abbas, M. (2023). Water Classification Using Convolutional Neural Network. IEEE Access.

6. Lehner, B., Liermann, C. R., Revenga, C., Vörösmarty, C., Fekete, B., Crouzet, P., Döll, P., Endejan, M., Frenken, K., Magome, J., Nilsson, C., Robertson, J. C., Rödel, R., Sindorf, N., and Wisser, D.: High-resolution mapping of the world's reservoirs and dams for sustainable river-flow management, Frontiers in Ecology and the Environment, 9, 494–502, https://doi.org/10.1890/100125, 2011.

7. Khazaei, B., Read, L. K., Casali, M., Sampson, K. M., and Yates, D. N.: GLOBathy, the global lakes bathymetry dataset, Sci Data, 9, 36, https://doi.org/10.1038/s41597-022-01132-9, 2022.

8. Schaperow, J. R., Li, D., Margulis, S. A., & Lettenmaier, D. P. (2019). A curve-fitting method for estimating bathymetry from water surface height and width. Water Resources Research, 55(5), 4288-4303.

9. Liu, K., Song, C., Wang, J., Ke, L., Zhu, Y., Zhu, J., ... & Luo, Z. (2020). Remote sensing-based modeling of the bathymetry and water storage for channel-type reservoirs worldwide. Water Resources Research, 56(11), e2020WR027147.

10. Li, Y., Zhao, G., Allen, G. H., and Gao, H.: Diminishing storage returns of reservoir construction, Nat Commun, 14, 3203, https://doi.org/10.1038/s41467-023-38843-5, 2023.

11. Vu, D. T., Dang, T. D., Galelli, S., and Hossain, F.: Satellite observations reveal 13 years of reservoir filling strategies, operating rules, and hydrological alterations in the Upper Mekong River basin, Hydrology and Earth System Sciences, 26, 2345–2364, https://doi.org/10.5194/hess-26-2345-2022, 2022.

---

## Author Response (AR2)

**Response to the topic editor:**

(1) One of my original comments was regarding how the limitations of using NDWI to classify water, such as issues related to high turbidity, shadows, the presence of aquatic vegetation, mixed land-water pixels, and seasonal vegetation cover, are addressed. In your response, you mention that the locally-adjusted Contrast Limited Adaptive Histogram Equalization (CLAHE, Reza, 2004) method can address all these issues. However, I think the authors should be cautious in asserting that CLAHE can effectively address all these challenges based on the method's underlying principle.

A: Yes, you are right. We want to clarify that while CLAHE applied to NDWI images does not specifically correct for high turbidity, shadows, aquatic vegetation, mixed land-water pixels, or seasonal vegetation effects—this remains a limitation of our study. However, CLAHE is intended to standardize reflectance values and reduce noise in NDWI-based water detection, helping to mitigate challenges such as varying illumination conditions and subtle spectral differences that could lead to partial misclassification of water pixels. We have now added a sentence in the revised manuscript to state these limitations explicitly.

Please also see Line 527-529:

"*Please note that CLAHE applied to NDWI images does not specifically correct for high turbidity, shadows, aquatic vegetation, mixed land-water pixels, or seasonal vegetation effects—this remains a limitation of our study.*"

(2) The temporal resolution of the reservoir storage time series is unclear. Given that this study leverages both Landsat and Sentinel-2 data, the temporal frequency should be higher than what is currently stated. It may be helpful to clarify this point in the time series figures to provide better explanation.

A: Thank you, we completely understand your concern. However, although we leveraged both Landsat and Sentinel-2 data and a robust cloud-effect removal algorithm, the estimation was still not possible for the images with more than 80% cloud coverage. We have also thought of combining a series of individual Landsat and Sentinel-2 images to generate surface water maps potentially up to a biweekly time period. However, many reservoirs do not fit in a single tile of Landsat and/or Sentinel-2, because of the shape and location of the reservoir. This leads to many no-data (missing) pixels over the reservoir, making image enhancement and gap-filling more challenging, and sometimes unrealistic. Therefore, we decided not to go for individual tiles of Landsat and Sentinel-2. Rather, we compromised slightly with the temporal frequency of the images and opted for using the image composites at 10-day intervals.

(3) Please increase the font in Figure 1, it's hard to read for now.

A: We have increased the font in Figure 1 to the best possible and updated the manuscript accordingly. Please see:

[Figure]

**Figure 1: Spatial distribution and evolution of reservoirs in Mainland Southeast Asia. (a) Map showing reservoir storage volume (km³), where the size of the circle is proportional to the reservoir capacity while the colour represents the year of commission of the reservoirs. (b) Basin-wise distribution of reservoir location (red dots), stream network in the respective catchments, and stream order. (c) Number of reservoirs built per year and their corresponding cumulative storage capacity. (d) Basin-wise total number of reservoirs built until 2023.**

(4) Statement 'Water frequency' is not very accurate in Figure 3, maybe 'Flood frequency'.

A: The term 'water frequency' here refers to how often water is present in a given pixel (area) within the reservoir, as explicitly stated in Lines 214-216. Since 'flood frequency' has a distinct meaning in hydrology—one that is less relevant to the reservoir's water surface dynamics—we believe 'water frequency' is the more appropriate term to describe the probability of water presence in the reservoir.

Lines 214-216:

"*The FREQ layer is created by making a composite of all binary NDWI images (more than 200 images from the Landsat and Sentinel collections), whose cloud percentage is less than 20% (i.e., clear sky condition) and by dividing it by the total number of selected images (cloud percentage <20%). We multiply the FREQ layer by 100 to get the percentage of water present at each pixel.*"